# Enhanced TARP-γ8-PSD-95 coupling in excitatory neurons contributes to the rapid antidepressant-like action of ketamine in male mice

Shi-Ge Xue [1,5], Jin-Gang He [1,2,3,4,5], Ling-Li Lu[1], Shi-Jie Song[1], Mei-Mei Chen[1], Fang Wang [1,2,3,4] ✉ & Jian-Guo Chen [1,2,3,4] ✉

Ketamine produces rapid antidepressant effects at sub-anesthetic dosage through early and sustained activation of α-amino-3-hydroxy-5-methyl-4-isoxazolepropionic acid receptors (AMPARs), however, the exact molecular mechanism still remains unclear. Transmembrane AMPAR regulatory protein-γ8 (TARP-γ8) is identified as one of AMPAR auxiliary subunits, which controls assemblies, surface trafficking and gating of AMPARs. Here, we show that ketamine rescues both depressive-like behaviors and the decreased AMPARs-mediated neurotransmission by recruitment of TARP-γ8 at the postsynaptic sites in the ventral hippocampus of stressed male mice. Furthermore, the rapid antidepressant effects of ketamine are abolished by selective blockade of TARP-γ8-containing AMPAR or uncoupling of TARP-γ8 from PSD-95. Overexpression of TARP-γ8 reverses chronic stress-induced depressive-like behaviors and attenuation of AMPARs-mediated neurotransmission. Conversely, knockdown of TARP-γ8 in excitatory neurons prevents the rapid antidepressant effects of ketamine.

Major depression disorder (MDD) is a common mood disorder, which is a leading cause of disability worldwide and one of the most burdensome illnesses globally with significant negative impacts on quality of life. Although selective serotonin reuptake inhibitors are currently used as the first-line antidepressants in the treatment of MDD, they have a delayed clinical onset within weeks to months, which dampens the general effect of treatment. Ketamine, a non-competitive *N*-methyl-*D*-aspartate receptor (NMDAR) antagonist, has been recently found to produce rapid antidepressant effects in patients with treatment-resistant depression (TRD) at sub-anesthetic dosage. Notably, accumulated evidence demonstrates that the antidepressant

effects of ketamine involve the activation of α-amino-3-hydroxy-5-methyl-4-isoxazolepropionic acid receptors (AMPARs)[1–3]. Ketamine's effects can be abolished by AMPAR antagonists and mimicked by the AMPARs-positive modulators[3–5]. The approval of esketamine for TRD and the involvement of AMPAR in the antidepressant effects of ketamine facilitate the idea that glutamatergic modulators can be considered as promising candidates for developing rapid-acting antidepressants[5,6]. Although studies have shown that AMPARs-positive allosteric modulators (AMPARs potentiators or AMPAkines) exert antidepressant effects, few direct AMPAR modulators have succeeded in early-stage clinical development[5], indicating that the direct

[1]State Key Laboratory for Diagnosis and Treatment of Severe Zoonotic Infectious Diseases, Department of Pharmacology, School of Basic Medicine, Tongji Medical College, Huazhong University of Science and Technology, Wuhan, China. [2]The Key Laboratory for Drug Target Researches and Pharmacodynamic Evaluation of Hubei Province, Wuhan, China. [3]The Research Center for Depression, Tongji Medical College, Huazhong University of Science, 430030 Wuhan, China. [4]Key Laboratory of Neurological Diseases (HUST), Ministry of Education of China, Wuhan, China. [5]These authors contributed equally: Shi-Ge Xue, Jin-Gang He. ✉e-mail: wangfanghust@hust.edu.cn; chenj@mails.tjmu.edu.cn

activation of AMPARs is not enough to fully elucidate the mechanism underlying rapid antidepressant effect of ketamine. In recent years, extensive studies have demonstrated that glutamatergic dysfunction is involved in the pathogenesis of stress-related mental illnesses, including MDD[7–9]. For example, chronic stress increases Neurensin-2 to facilitate endocytosis of AMPARs in the hippocampus. On the contrary, knockout of Neurensin-2 mediates resilience to stress and enhances synaptic anchoring of AMPARs[10]. In addition, AMPARs-mediated excitatory synaptic transmission is decreased in the prefrontal cortex and hippocampus after stress, which can be reversed by antidepressant treatment[9–11]. Postmortem investigations also indicate that a disruption of the glutamate system occurs in MDD patients who complete suicide[12,13].

Functional AMPARs are assembled from four core subunits GluA1-GluA4 together with auxiliary subunits. Transmembrane AMPA receptor regulatory proteins (TARPs) are identified as the first family of auxiliary subunits of ionotropic receptors that control AMPAR trafficking, channel activity and pharmacology. According to the distinct functions in AMPAR modulation, TARPs are classified into two subfamilies: Class I TARPs that contain a typical PDZ domain-binding motif (-TTPV) at the C terminus, including stargazin/γ2, γ3, γ4, and γ8; Class II TARPs include γ5 and γ7[14]. TARP-γ8 encoded by *Cacng8* gene is the most abundant TARP in hippocampal neurons[15]. TARP-γ8 interacts with postsynaptic density-95 (PSD-95) via PDZ domain to control AMPAR trafficking and long-term potentiation[16]. The observation that the synaptic AMPAR function is severely impaired in TARP-γ8 knockout mice indicates that TARP-γ8 is necessary for the basal AMPAR expression and localization in the hippocampus[17]. Another previous research suggests that TARP-γ8 in the ventral hippocampus is required for calcium/calmodulin-dependent protein kinase II (CaMKII)β-mediated stress resilience[18]. Taken together, converging results suggest that ketamine produces its antidepressant effects through enhancing glutamatergic neurotransmission in hippocampal pyramidal neurons, however, the molecular mechanism remains unclear. Here, we wondered whether TARP-γ8-mediated regulation of AMPAR was involved in the effects of ketamine.

In the present study, we found that ketamine increased the phosphorylation of CaMKII and the binding of TARP-γ8 with PSD-95 in the ventral hippocampus. Blockade of TARP-γ8-selective AMPAR activity in the ventral hippocampus abolished the antidepressant effects induced by ketamine. Moreover, knockdown of TARP-γ8 in the excitatory neurons of the ventral hippocampus mimicked the depressive-like phenotypes and prevented the rapid antidepressant effect of ketamine. These results suggest that TARP-γ8 is the potential molecular target of ketamine.

## Results

### Ketamine induces recruitment of TARP-γ8 at the postsynaptic sites in ventral hippocampus of male stressed mice

Strong evidence supports the rapid antidepressant effect of a single intravenous ketamine infusion for the patient with MDD. Considering that the antidepressant action of ketamine is mediated through the modulation of AMPARs, and TARP-γ8 regulates the trafficking and gating properties of AMPARs, we wondered whether TARP-γ8 was involved in the effect of ketamine. First, we employed chronic social defeat stress (CSDS) model to induce depressive-like phenotypes in mice[19]. Male mice were exposed to chronic defeat stress for 10 consecutive days (Supplementary Fig. 1a). After CSDS paradigm, the social interaction test (SIT), sucrose preference test (SPT), forced swimming test (FST) and open field test (OFT) were carried out. Compared with mice in control group (CON), the stressed mice showed social avoidance in the SIT, anhedonia in the SPT and despair-like behaviors in the FST, but without changes on locomotor activity (Supplementary Fig. 1b–h). The reduced social interaction ratio, time spent in the interaction area when the target was present, and the sucrose

preference of stressed mice were rescued 2 h (Fig. 1a–c) and 24 h (Supplementary Fig. 2a–c) after a single dose of ketamine (10 mg/kg, i.p.). Meanwhile, ketamine treatment reduced the immobility time in stressed mice in the FST without significant effect on general locomotor activity (Fig. 1d, e and Supplementary Fig. 2d, e). It has been demonstrated that the ventral hippocampus is responsible for emotional processing and antidepressant action of ketamine[20–22]. To assess whether ketamine exerts its antidepressant effects through the ventral hippocampus, we performed local infusion of ketamine (0.2, 2 or 20 μg, bilaterally) into the ventral hippocampus of male mice. We observed significant antidepressant effects 1 h and 24 h after a single intra-ventral hippocampus injection of ketamine (2 μg each side, bilaterally) (Supplementary Fig. 3a–f).

Growing evidence indicates that AMPARs activation is involved in the rapid antidepressant effects of ketamine in animal models of depression. To test whether AMPARs-mediated excitatory transmission in the ventral hippocampus was modulated by chronic stress and ketamine, we performed whole-cell patch-clamp recording on CA1 neurons in ventral hippocampal slices. Compared with that of control group, the amplitude but not frequency of mEPSC was reduced in the CA1 neurons of CSDS-treated mice (Fig. 1f, g). However, bath incubation of ketamine (10 μM) for 1 h enhanced the amplitude of AMPARs-mediated mEPSC in both neurons from control and stressed group, without effect on the frequency, the rise time and the decay time of AMPARs-mediated mEPSC (Fig. 1g and Supplementary Fig. 4a, b). These results indicate that chronic stress reduces AMPARs-mediated synaptic transmission in the ventral hippocampus, which is reversed by ketamine.

The increased amplitude of AMPARs-mediated mEPSC in combination with a lack of effect on the frequency suggests that the effects of ketamine on synaptic transmission involve postsynaptic aspects of glutamatergic transmission. Previous studies have shown that TARP-γ8 is proposed to enhance the surface expression of AMPARs and stabilize AMPARs at synaptic site by directly interacting with PSD-95[16]. We hypothesized that ketamine-driven enhancement of AMPAR-mediated excitatory synaptic transmission might be related to synaptic anchoring of TARP-γ8 in the ventral hippocampus. Co-immunoprecipitation assay was used to determine the influence of ketamine on the interaction between TARP-γ8 and PSD-95 in vitro and in vivo. We observed that the binding of TARP-γ8 to PSD-95 was increased at 1 h and 24 h after ketamine (10 μM) treatment in primary cultured hippocampal neurons (Fig. 1h, i). Meanwhile, the total expression of PSD-95 and TARP-γ8 was not affected by ketamine (Fig. 1j, k). Furthermore, the ventral hippocampal brain tissue was obtained 2 h after intraperitoneal injection of ketamine (10 mg/kg, i.p.) and we observed that the binding of TARP-γ8 with PSD-95 was inversely modulated by chronic stress and ketamine in vivo. Ketamine increased the interaction between TARP-γ8 and PSD-95 in control mice and rescued the disruption of TARP-γ8-PSD-95 interaction induced by CSDS (Fig. 1l, m). The total expression of PSD-95 was not affected by neither chronic stress nor ketamine (Fig. 1n). Moreover, we found that the total expression of TARP-γ8 in the ventral hippocampus was downregulated by CSDS, which was not affected by ketamine (Fig. 1o). The above results suggest that ketamine rescues the dysfunction of AMPARs-mediated synaptic transmission induced by chronic stress though enhancing recruitment of TARP-γ8 at the postsynaptic sites.

### Potentiation of TARP-γ8-selective AMPAR-mediated synaptic transmission in the ventral hippocampus is crucial for the antidepressant effects of ketamine

As the predominant TARP throughout the hippocampus, TARP-γ8 is particularly attractive in developing subtype-selective AMPAR modulator. JNJ55511118 (Fig. 2a), is a specific negative modulator of AMPAR containing TARP-γ8 (TARP-γ8-selective AMPAR), but not other TARPs in the hippocampus[23,24]. To further investigate the role of TARP-γ8-

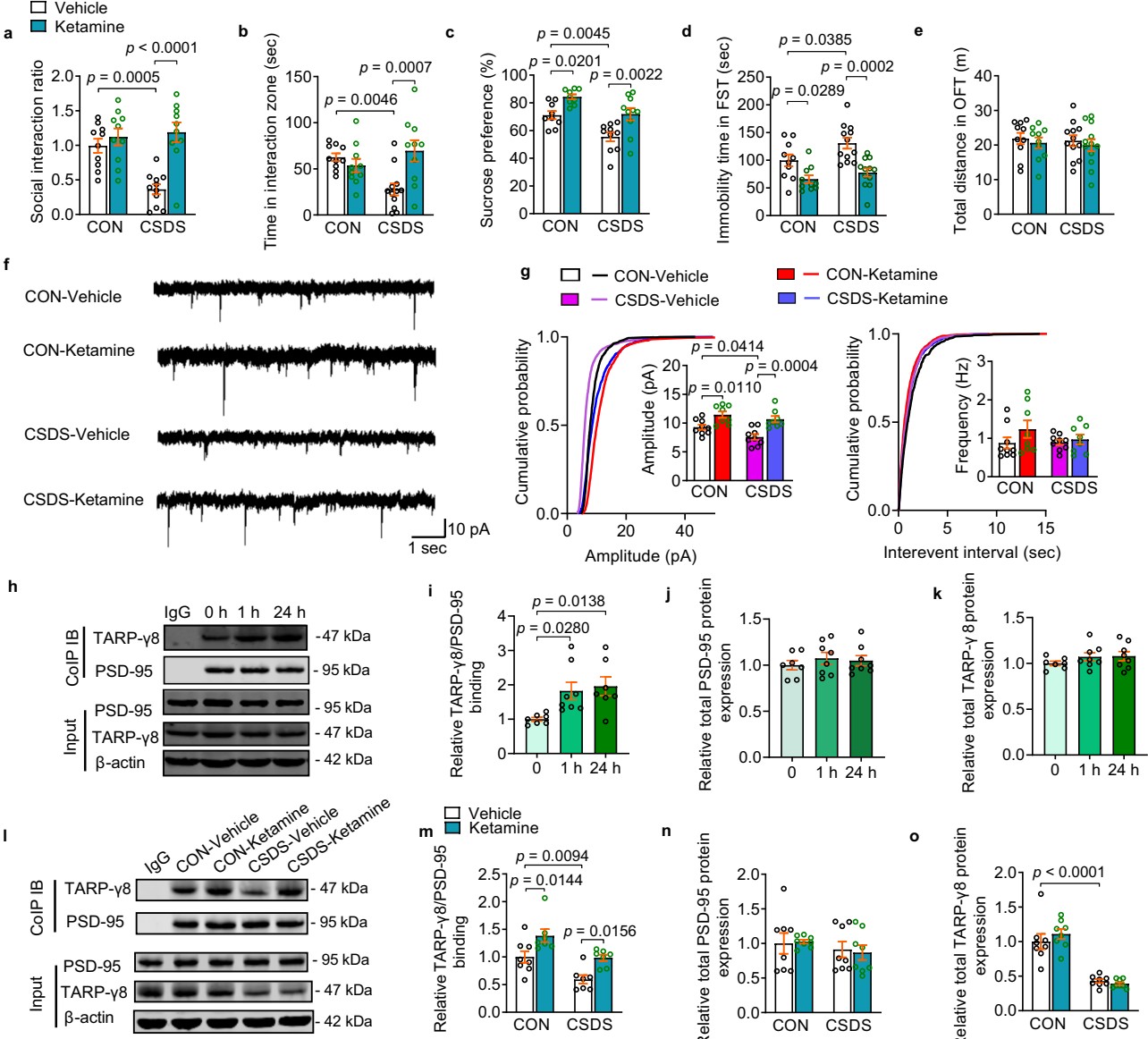

**Fig. 1 | Ketamine drives recruitment of TARP-γ8 at the postsynaptic sites in chronic stress model. a** Social interaction ratio of SIT (*n* = 10, 10, 11, and 10 in CON-Vehicle, CON-Ketamine, CSDS-Vehicle and CSDS-Ketamine, respectively). **b** Time in the interaction zone of SIT (*n* = 10, 10, 12, and 10 in CON-Vehicle, CON-Ketamine, CSDS-Vehicle and CSDS-Ketamine, respectively). **c** Preference for sucrose in the SPT (*n* = 9, 9, 10, and 10 in CON-Vehicle, CON-Ketamine, CSDS-Vehicle and CSDS-Ketamine, respectively). **d, e** Immobility time in the FST (**d**) and total distance in the OFT (**e**) (*n* = 10, 10, 12, and 12 in CON-Vehicle, CON-Ketamine, CSDS-Vehicle and CSDS-Ketamine, respectively). **f** Representative traces of AMPARs-mediated mEPSC recordings in ventral hippocampal CA1 neurons. **g** Quantification of cumulative probability and amplitude and frequency of mEPSCs (*n* = 9 cells from 4 mice in CON-Vehicle, *n* = 8 cells from 3 mice in CON-Ketamine, *n* = 9 cells from 3 mice in CSDS-Vehicle, *n* = 8 cells from 4 mice in CSDS-Ketamine, respectively). **h** Representative

western blot image of Co-IP assay. **i** Quantification of association between PSD-95 and TARP-γ8 in cultured primary hippocampal neurons (*n* = 7, 8, and 7 in 0, 1 h and 24 h, respectively). **j, k** Quantification of total expression of PSD-95 (**j**) and TARP-γ8 (**k**) in cultured primary hippocampal neurons (*n* = 7, 8, and 8 in 0, 1 h, and 24 h, respectively). **l** Representative western blot image of Co-IP assay. **m** Quantification of association between PSD-95 and TARP-γ8 in the ventral hippocampus (*n* = 8, 7, 7, and 7 in CON-Vehicle, CON-Ketamine, CSDS-Vehicle and CSDS-Ketamine, respectively). **n, o** Quantification of total expression of PSD-95 (**n**) and TARP-γ8 (**o**) in the ventral hippocampus (*n* = 8 per group). Comparisons were performed by two-way ANOVA analysis followed by Bonferroni's multiple comparisons test in (**a**–**g**, **m**–**o**) and one-way ANOVA analysis followed by Dunnett's multiple comparisons test in **i**–**k**. Data are presented as mean ± SEM. All numbers (*n*) are biologically independent experiments. Source data are provided as a Source Data file.

mediated regulation of AMPAR in the rapid antidepressant effects of ketamine, JNJ55511118 was employed. We first screened the effective concentration in the FST by intra-ventral hippocampus injection of JNJ55511118 bilaterally at the dosage of 1 µM and 10 µM, and then evaluated behavioral changes after treatment for 30 min and 24 h. We found that intra-ventral hippocampus injection of 10 µM, but not 1 µM JNJ55511118 induced reduced preference for sucrose in the SPT and an increase in immobility time in the FST, without effect on locomotor activity in the OFT at both 30 min (Fig. 2b–d) and 24 h (Supplementary

Fig. 5a–c), indicating inhibition of TARP-γ8-selective AMPAR induces depressive-like behaviors of mice. JNJ55511118 (10 µM) was further infused into the ventral hippocampus bilaterally 30 min before ketamine administration (10 mg/kg, i.p.) in CSDS-exposed mice (Fig. 2e, f). It was shown that pre-administration of JNJ55511118 into the ventral hippocampus abolished the antidepressant effects of ketamine in CSDS model, without influence on locomotor activity 2 h and 24 h after ketamine administration (Fig. 2g–j and Supplementary Fig. 6a–d). Together, these results indicate that pharmacological blockade of

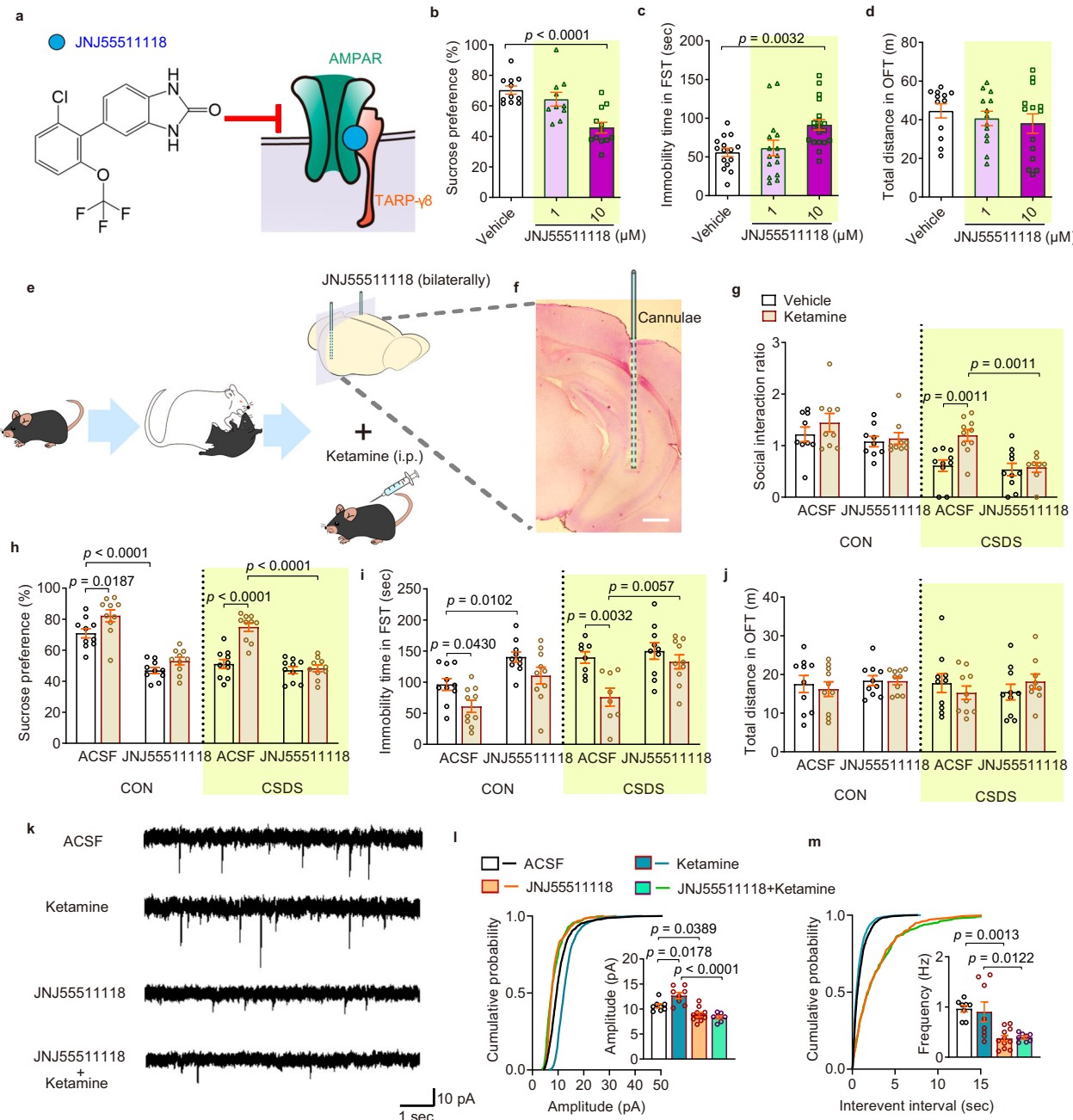

**Fig. 2 | Inhibition of TARP-γ8-selective AMPARs abolishes the acute effects of ketamine in the ventral hippocampus. a** Structures of the mechanism of JNJ55511118. **b** Preference for sucrose in the SPT ($n$ = 11, 10, and 12 in Vehicle, 1 μM and 10 μM, respectively).**c** Immobility time in the FST ($n$ = 15, 15, and 19 in Vehicle, 1 μM, and 10 μM, respectively). **d** Total distance in the OFT ($n$ = 12, 13, and 14 in Vehicle, 1 μM, and 10 μM, respectively). **e, f** Schematic diagram of experiment (**e**) and histological verification of the cannula placement in the ventral hippocampus with Nissl staining (**f**). Scale bar = 500 μm. **g** Social interaction ratio of SIT ($n$ = 9, 9, 9, 9, 10, 10, 10, and 8 in CON-ACSF-Vehicle, CON-ACSF-Ketamine, CON-JNJ55511118-Vehicle, CON-JNJ55511118-Ketamine, CSDS-ACSF-Vehicle, CSDS-ACSF-Ketamine, CSDS-JNJ55511118-Vehicle and CSDS-JNJ55511118-Ketamine, respectively). **h** Preference for sucrose in the SPT ($n$ = 10 mice per group). **i** Immobility time in the FST ($n$ = 10, 10, 10, 10, 8, 8, 10 and 10 in CON-ACSF-Vehicle, CON-ACSF-Ketamine, CON-JNJ55511118-Vehicle, CON-JNJ55511118-Ketamine, CSDS-ACSF-Vehicle, CSDS-ACSF-

Ketamine, CSDS-JNJ55511118-Vehicle and CSDS-JNJ55511118-Ketamine, respectively). **j** Total distance traveled in the OFT ($n$ = 10, 10, 10, 10, 10, 10, 10 and 9 in CON-ACSF-Vehicle, CON-ACSF-Ketamine, CON-JNJ55511118-Vehicle, CON-JNJ55511118-Ketamine, CSDS-ACSF-Vehicle, CSDS-ACSF-Ketamine, CSDS-JNJ55511118-Vehicle and CSDS-JNJ55511118-Ketamine, respectively). **k** Representative traces of AMPARs-mediated mEPSC recordings in ventral hippocampal CA1 neurons. **l, m** Quantification of cumulative probability and amplitude (**l**) and frequency (**m**) of mEPSCs ($n$ = 8 cells from 4 mice in ACSF, $n$ = 8 cells from 4 mice in Ketamine, $n$ = 11 cells from 5 mice in JNJ55511118, $n$ = 8 cells from four mice in JNJ55511118-Ketamine). Comparisons were performed by one-way ANOVA analysis followed by Dunnett's multiple comparisons test in **b–d**, by two-way and one-way ANOVA analysis followed by Bonferroni's multiple comparisons test in (**g–j**) and in (**l, m**), respectively. Data are presented as mean ± SEM. All numbers ($n$) are biologically independent experiments. Source data are provided as a Source Data file.

TARP-γ8-selective AMPAR in the ventral hippocampus is sufficient to abolish the rapid antidepressant effect of ketamine.

To further validate the role of TARP-γ8-selective AMPAR in electrophysiological effects of ketamine in the ventral hippocampus, we employed whole-cell patch-clamp recording. We observed that bath incubation of JNJ55511118 (1 μM) for 30 min impaired both amplitude and frequency of mEPSC in ventral hippocampal CA1 neurons (Fig. 2k–m). Then, bath incubation of JNJ55511118 (1 μM) was administered before ketamine application (10 μM) in electrophysiological experiment. The results showed that the enhanced amplitude of mEPSCs induced by bath incubation of ketamine was blocked by pretreatment with JNJ55511118 in the ventral hippocampal slices (Fig. 2l). Previous study has revealed that TARP-γ8 is involved in regulating the kinetic properties of AMPAR. Our results showed that AMPAR kinetics, such as decay time was reduced by JNJ55511118 rather than ketamine (Supplementary Fig. 7a, b). The promoting effect of JNJ55511118 on the desensitization of AMPARs was also consistent with a recent report, which identified that JNJ55511118 decreased peak currents and the time constant of desensitization of AMPARs[25]. Taken together, these results provide important insights into the rapid antidepressant effects of ketamine that is dependent on the potentiation of TARP-γ8-selective AMPAR-mediated synaptic transmission in the ventral hippocampus.

## Overexpression of postsynaptic TARP-γ8 in the ventral hippocampus reverses the cellular and behavioral changes in stressed mice

Although CSDS reduced the interaction between TARP-γ8 and PSD-95 and the total expression of TARP-γ8 in the ventral hippocampus, it was not clear whether the deficiency of TARP-γ8/PSD-95 complex was correlated with CSDS-induced depressive-like behaviors. TARP-γ8 has been demonstrated to interact with PSD-95 via C-terminal-TTPV motif, which is required for recruitment of TARP-γ8-containing AMPAR complexes into synapses. Therefore, we constructed lentiviral vectors (LV) to express the full-length TARP-γ8, or TARP-γ8-Δ4 (lacking the -TTPV PDZ domain) in the bilateral ventral hippocampus of mice (Fig. 3a, b). Two weeks after the virus-injection (2 μl, each side), we found that overexpression of the full-length TARP-γ8, but not TARP-γ8-Δ4, reversed the social avoidance and the despair behaviors of stressed mice (Fig. 3c–e). Meanwhile, the locomotor activity of mice remained unchanged (Fig. 3f). These results suggest that stress-induced deficiency of TARP-γ8 at the postsynaptic sites drives depressive-like behaviors.

Previous studies have shown that the surface expressions of GluA1 and GluA2 are reduced in the hippocampus of stressed mice induced by CSDS[26]. Here, we found that although overexpression of both TARP-γ8 and TARP-γ8-Δ4 increased the expression of TARP-γ8 protein in stressed mice, the decreased expression of surface GluA1 and GluA2 proteins induced by CSDS were rescued by overexpression of TARP-γ8 rather than TARP-γ8-Δ4 in the ventral hippocampus (Fig. 3g, h), whereas the total amounts of AMPAR subunits were not affected (Supplementary Fig. 8a, b). These results indicate that the binding of TARP-γ8/PSD-95 plays a key role in the antidepressant effects under stress condition.

Next, the effect of interaction between TARP-γ8 and PSD-95 on AMPARs-mediated mEPSC was investigated under stressed condition by using whole-cell patch-clamp recording. Overexpression of full-length TARP-γ8 rather than TARP-γ8-Δ4 reversed CSDS-induced decrease in amplitude of AMPARs-mediated mEPSC in the ventral hippocampal CA1 neuron, without effect on the frequency and kinetic properties of mEPSC, such as rise and decay time (Fig. 3i–m and Supplementary Fig. 8c, d). The results indicate that overexpression of postsynaptic TARP-γ8 exhibits antidepressant action via maintaining AMPARs-mediated synaptic transmission in the ventral hippocampus during stress.

## Knockdown of TARP-γ8 in the ventral hippocampus induces depressive-like behaviors and reduces AMPARs-mediated synaptic transmission

Considering that the loss of TARP-γ8 in the ventral hippocampus was observed in the stressed mice, we asked whether the deficiency of TARP-γ8 in the ventral hippocampus contributed to the depressive-like behaviors of mice. First, we constructed adeno-associated virus (AAV) vector containing short hairpin RNA (shRNA) targeted for TARP-γ8 (AAV-*Cacng8*-shRNA) to achieve knockdown of TARP-γ8 (Fig. 4a, b). After injection of AAV vector into the ventral hippocampus (2 μl, each side) for 28 d, we found that AAV-*Cacng8* shRNA reduced the protein expression of TARP-γ8 by ~60% (Fig. 4c). Behavioral results revealed that knockdown of TARP-γ8 in the ventral hippocampus increased immobility time in the tail suspension test (TST) and FST (Fig. 4d, e). Consistent with a previous report that TARP-γ8 knockout mice were hyperactive[27], here we noticed that mice lacking TARP-γ8 in the ventral hippocampus displayed increased locomotor activity in the OFT (Fig. 4f). The results indicate that the lack of TARP-γ8 in the ventral hippocampus drives depressive-like behaviors similar to chronic stress.

Auxiliary subunits control almost all aspects of AMPAR function in the brain, such as conductance, subunit composition and trafficking[28]. TARPs are principal AMPAR-auxiliary subunits, in which TARP-γ8 is predominantly expressed in the hippocampus[15]. However, the impact of specific knockdown of TARP-γ8 in the ventral hippocampus on the AMPARs function is not clear. We next quantified the surface expression of AMPAR subunit GluA1 and GluA2 and found that both the total expressions of GluA1 and GluA2 and surface GluA2 were decreased in the ventral hippocampus of TARP-γ8-knockdown mice (Fig. 4g, h). Considering that AMPARs are fundamental elements in excitatory neurotransmission, the expression of AMPAR subunits in the synaptic compartment after knockdown of TARP-γ8 was further measured. It was shown that both GluA1 and GluA2 expression were decreased in the synaptic fraction (Supplementary Fig. 9a, b). The electrophysiological results also revealed that the amplitude of AMPARs-mediated mEPSCs was decreased after knockdown of TARP-γ8, whereas the frequency was unchanged (Fig. 4i–k). Other kinetic properties of mEPSCs such as decay time was reduced, but the rise time was not changed (Supplementary Fig. 9c, d). These results suggest that loss of TARP-γ8 in the ventral hippocampus impairs the composition and synaptic localization of AMPARs.

## The antidepressant effect of ketamine requires CaMKIIα-dependent TARP-γ8-PSD-95 coupling

It has been reported that acute ketamine administration stimulates the phosphorylation of CaMKII in vitro and in vivo[29,30]. The C-terminal of TARP-γ8 contains several serine residues, phosphorylation of TARP C-terminal by CaMKII allows the binding to PSD-95 and synaptic localization of receptors[31–33]. We thus hypothesized that the increased phosphorylation of CaMKII in the ventral hippocampus mediated the rapid antidepressant effects of ketamine. Western blot results showed that acute ketamine treatment increased the level of p-CaMKIIα in the ventral hippocampus (Fig. 5a, b). In addition, intra-ventral hippocampus injection of CaMKII inhibitor KN93 (10 μM) abolished the rapid antidepressant action of ketamine in the SPT and FST (Fig. 5c, d), without alteration on locomotion activity (Fig. 5e), which was consistent with previous report[30]. Furthermore, we found that KN93 prevented ketamine-induced increase in the binding of TARP-γ8 and PSD-95 (Fig. 5f, g), however, the total expressions of PSD-95 and TARP-γ8 proteins were not affected by KN93 (Fig. 5h, i), suggesting that ketamine increases the binding of TARP-γ8 and PSD-95 through CaMKII phosphorylation. Considering that the increased binding of TARP-γ8 and PSD-95 contributed to the enhanced excitatory synaptic transmission mediated by AMPAR, electrophysiological recording was conducted to further elucidate the role of CaMKII phosphorylation in

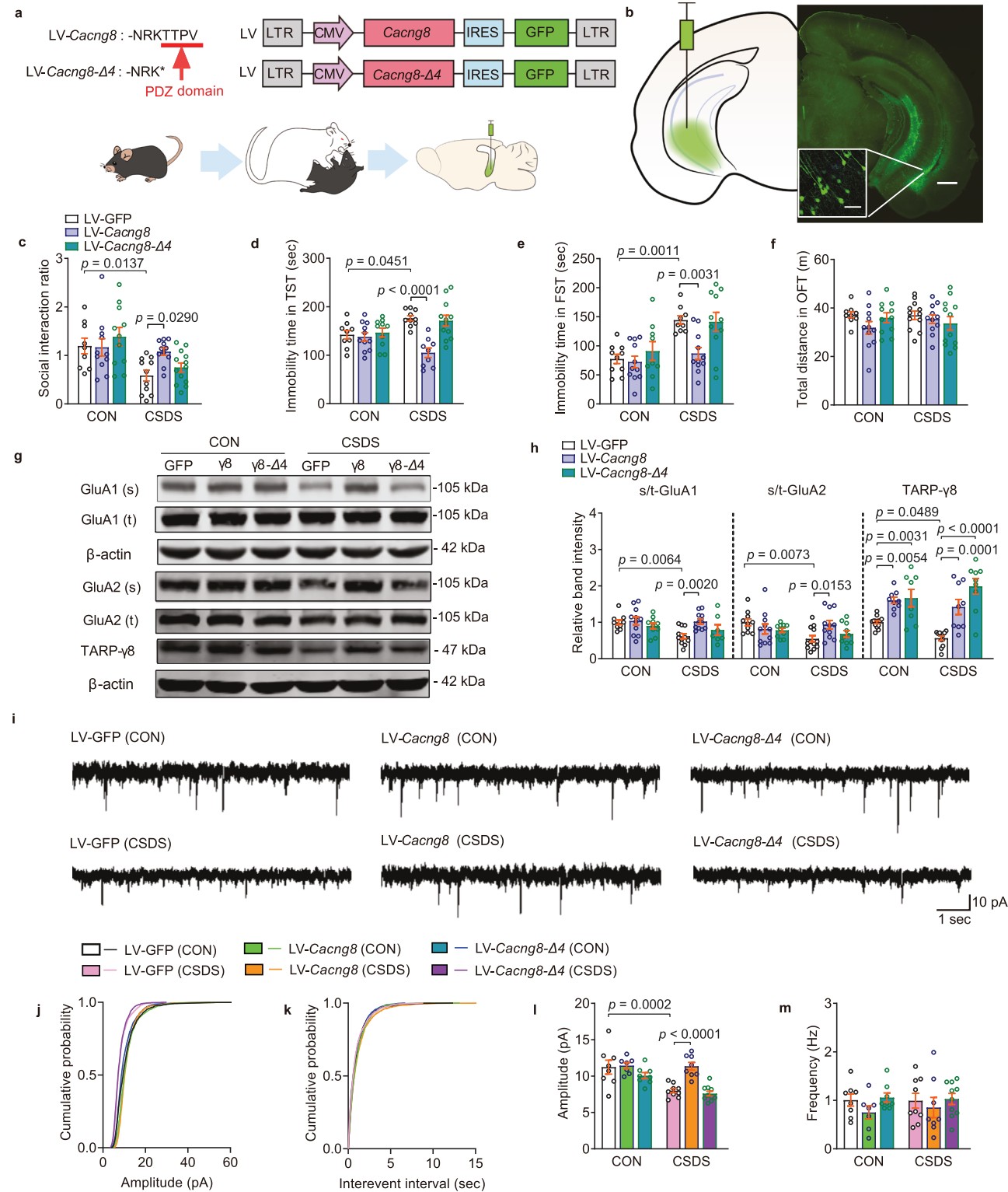

ketamine-induced enhancement of excitatory synaptic transmission. The results showed that pre-incubation of KN93 for 30 min prevented the enhanced amplitude of mEPSCs in the ventral hippocampal slices induced by ketamine (Fig. 5j, k), without effect on the frequency and kinetic properties of mEPSCs, such as rise and decay time (Fig. 5l and Supplementary Fig. 10a, b), indicating that ketamine promotes post-synaptic localization of TARP-γ8 and AMPARs-mediated synaptic transmission in an CaMKII phosphorylation-dependent manner.

Next, we asked whether uncoupling of TARP-γ8 from PSD-95 prevented the behavioral effects of ketamine. A Tat-TTPV peptide was

designed, in which the TTPV motif at PDZ domain of TARP-γ8 (LNRKTTPV) was fused to the cell permeable peptide Tat. The efficacy of Tat-TTPV was evaluated in vivo, and we found that intra-ventral hippocampus injection of Tat-TTPV (10 μM) disrupted the TARP-γ8/PSD-95 interaction (Fig. 5m, n and Supplementary Fig. 11a, b). Then, intra-ventral hippocampus injection of Tat-TTPV was performed 30 min before ketamine administration (10 mg/kg, i.p.), followed by behavioral tests. It was shown that Tat-TTPV treatment abolished the rapid antidepressant action of ketamine in the SPT and FST (Fig. 5o, p) without alteration on locomotion activity (Fig. 5q), indicating the

**Fig. 3 | Overexpression of TARP-γ8 but not TARP-γ8-Δ4 in the ventral hippocampus prevents CSDS-induced synaptic and behavioral impairment.**
**a** Schematic representation of LV-mediated TARP-γ8 or TARP-γ8-Δ4 overexpression and a schematic diagram of the experimental process. **b** Targeted locations and confocal images of GFP (green) expression in the ventral hippocampus. Scale bar = 50 μm (left), 500 μm (right). Experiments were repeated independently 3 times with similar results. **c** Social interaction ratio of SIT (*n* = 9, 11, 11, 11, 11, and 13 in CON-LV-GFP, CSDS-LV-GFP, CON-LV-*Cacng8*, CSDS-LV-*Cacng8*, CON-LV-*Cacng8-Δ4* and CSDS-LV-*Cacng8-Δ4*, respectively). **d** Immobility time in the TST (*n* = 9, 10, 11, 9, 10, and 12 in CON-LV-GFP, CSDS-LV-GFP, CON-LV-*Cacng8*, CSDS-LV-*Cacng8*, CON-LV-*Cacng8-Δ4* and CSDS-LV-*Cacng8-Δ4*, respectively). **e** Immobility time in the FST (*n* = 10, 9, 11, 11, 9, and 11 in CON-LV-GFP, CSDS-LV-GFP, CON-LV-*Cacng8*, CSDS-LV-*Cacng8*, CON-LV-*Cacng8-Δ4* and CSDS-LV-*Cacng8-Δ4*, respectively). **f** Total distance in the OFT (*n* = 9, 12, 11, 11, 11, and 12 in CON-LV-GFP, CSDS-LV-GFP, CON-LV-*Cacng8*, CSDS-LV-*Cacng8*, CON-LV-*Cacng8-Δ4* and CSDS-LV-*Cacng8-Δ4*, respectively). **g** The representative image of the western blot.

**h** Quantification of protein expression of surface GluA1 (*n* = 10, 12, 11, 11, 10, and 7 in CON-LV-GFP, CSDS-LV-GFP, CON-LV-*Cacng8*, CSDS-LV-*Cacng8*, CON-LV-*Cacng8-Δ4* and CSDS-LV-*Cacng8-Δ4*, respectively), surface GluA2 (*n* = 10, 12, 11, 11, 10, and 10 in CON-LV-GFP, CSDS-LV-GFP, CON-LV-*Cacng8*, CSDS-LV-*Cacng8*, CON-LV-*Cacng8-Δ4* and CSDS-LV-*Cacng8-Δ4*, respectively) and TARP-γ8 (*n* = 14, 12, 9, 9, 8, and 9 in CON-LV-GFP, CSDS-LV-GFP, CON-LV-*Cacng8*, CSDS-LV-*Cacng8*, CON-LV-*Cacng8-Δ4* and CSDS-LV-*Cacng8-Δ4*, respectively). **i** Representative traces of AMPARs-mediated mEPSC recordings from different groups. **j**–**m** Quantification of cumulative probability (**j**, **k**) and amplitude (**l**) and frequency (**m**) of mEPSCs (*n* = 8 cells from 5 mice in CON-LV-GFP, *n* = 9 cells from 6 mice in CSDS-LV-GFP, *n* = 7 cells from 6 mice in CON-LV-*Cacng8*, *n* = 8 cells from 5 mice in CSDS-LV-*Cacng8*, *n* = 8 cells from 6 mice in CON-LV-*Cacng8-Δ4* and *n* = 10 cells from 6 mice in CSDS-LV-*Cacng8-Δ4*). Comparisons were performed by two-way ANOVA analysis followed by Bonferroni's multiple comparisons test. s/t stands for surface/total in **h**. Data are presented as mean ± SEM. All numbers (*n*) are biologically independent experiments. Source data are provided as a Source Data file.

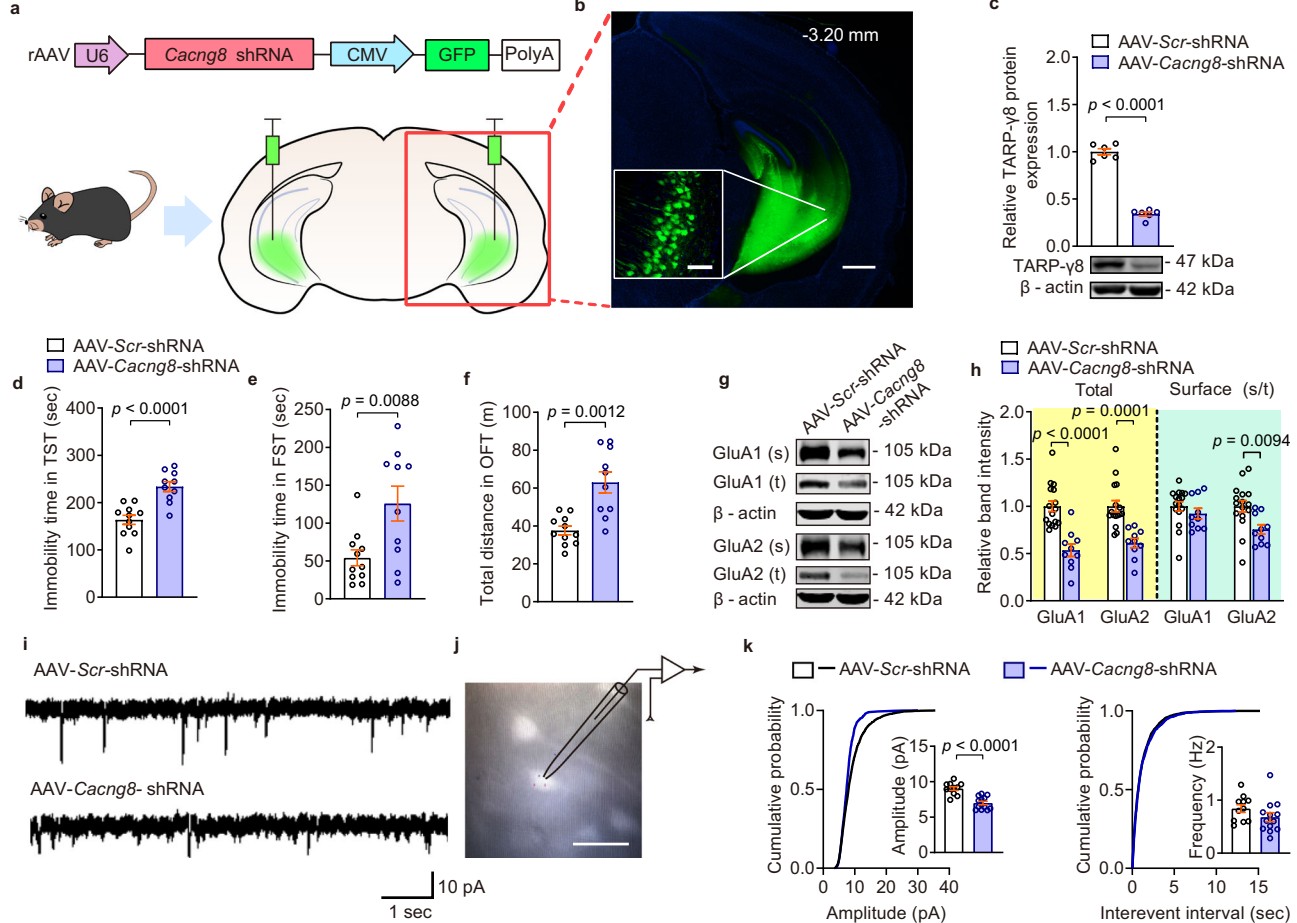

**Fig. 4 | Knockdown of TARP-γ8 in the ventral hippocampus induces depressive-like phenotypes in mice. a** AAV-expressing constructs encoding GFP and shRNAs targeting TARP-γ8 and region-specific expression of GFP in the ventral hippocampus of mice. **b** Targeted locations and confocal images of AAV-mediated GFP (green) expression in the ventral hippocampus. Scale bar = 100 μm (inside), 500 μm (outside). Experiments were repeated independently three times with similar results. **c** Quantification of TARP-γ8 protein expression (*n* = 6 mice per group). **d**, **e** Immobility time in the TST (**d**) and FST (**e**) (*n* = 11 mice in AAV-*Scr*-shRNA, *n* = 10 mice in AAV-*Cacng8*-shRNA). **f** Total distance traveled in the OFT (*n* = 11 mice in AAV-*Scr*-shRNA, *n* = 10 mice in AAV-*Cacng8*-shRNA). **g** The representative image of western blot. **h** The quantification of total and surface protein

expression of GluA1 and GluA2 (*n* = 16 mice in AAV-*Scr*-shRNA, *n* = 10 mice in AAV-*Cacng8*-shRNA). **i** Representative traces of AMPARs-mediated mEPSC recordings from different groups. **j** Whole-cell patch-clamp was used to record AMPARs-mediated mEPSC of CA1 neurons in the ventral hippocampus with GFP (bright) expression. Scale bar = 50 μm. **k** Quantification of cumulative probability and amplitude and frequency of mEPSCs (*n* = 11 cells from four mice in AAV-*Scr*-shRNA, *n* = 13 cells from 6 mice in AAV-*Cacng8*-shRNA). Comparisons were performed by unpaired, two-tailed *t* test. Data are presented as mean ± SEM. All numbers (*n*) are biologically independent experiments. Source data are provided as a Source Data file.

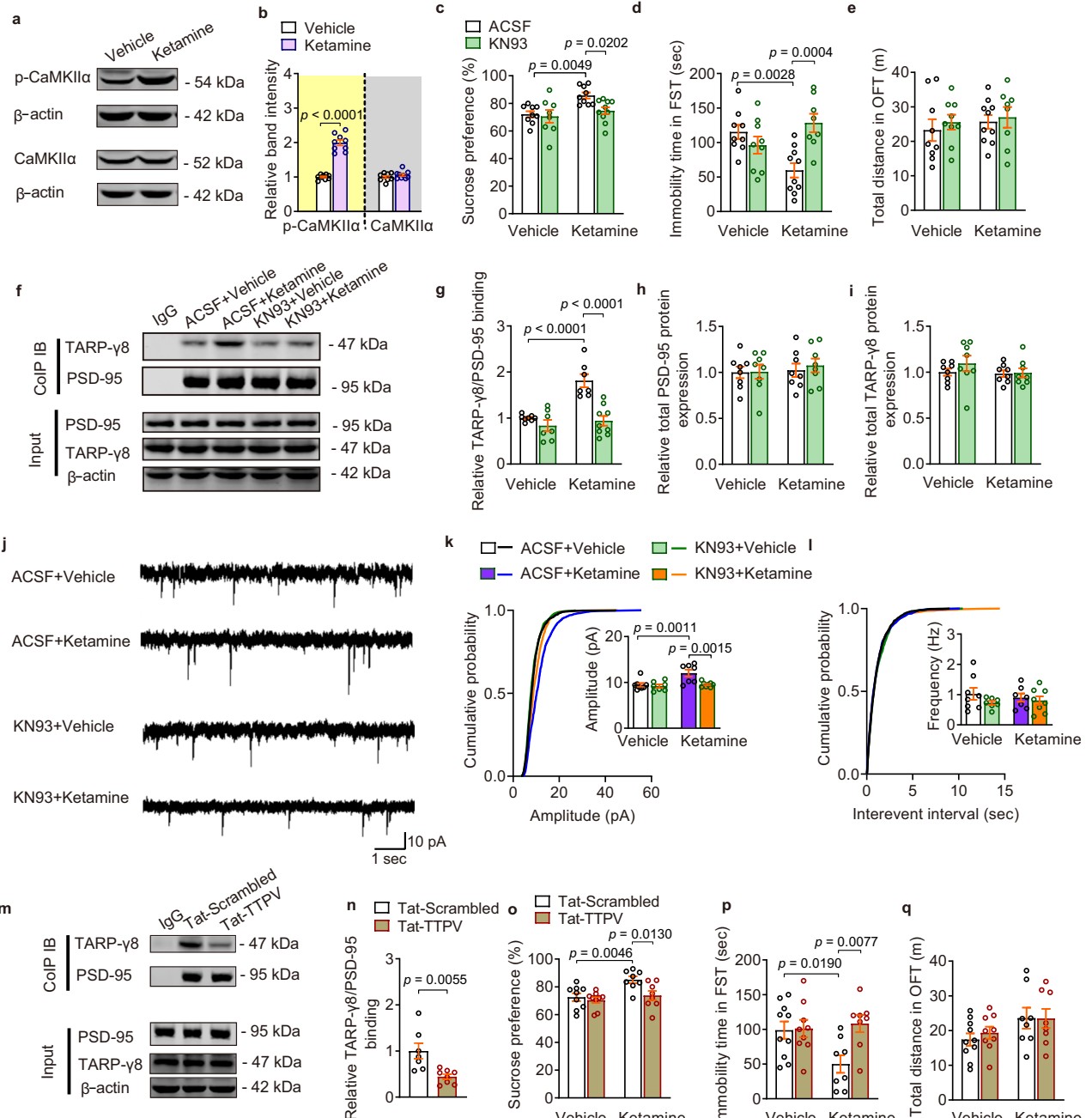

**Fig. 5 | CaMKIIα-dependent TARP-γ8-PSD-95 coupling is necessary for the rapid antidepressant effects of ketamine. a** The representative image of western blot. **b** The quantification of protein expression of p-CaMKIIα and CaMKIIα ($n = 9$ per group). **c** Preference for sucrose in the SPT ($n = 9, 8, 9$, and 10 in Vehicle-ACSF, Vehicle-KN93, Ketamine-ACSF, and Ketamine-KN93, respectively). **d, e** Immobility time in the FST (**d**) and total distance in the OFT (**e**) ($n = 9, 9, 10$, and 8 in Vehicle-ACSF, Vehicle-KN93, Ketamine-ACSF and Ketamine-KN93, respectively). **f** Representative western blot image of Co-IP assay. **g** Quantification of association between PSD-95 and TARP-γ8 ($n = 8, 7, 7$, and 9 in Vehicle-ACSF, Vehicle-KN93, Ketamine-ACSF and Ketamine-KN93, respectively). **h, i** Total protein expression of PSD-95 (**h**) and TARP-γ8 (**i**) ($n = 8$ samples per group). **j** Representative traces of AMPARs-mediated mEPSC recordings in ventral hippocampal CA1 neurons from different groups. **k, l** Quantification of cumulative probability and amplitude (**k**)

and frequency (**l**) of mEPSCs ($n = 8$ cells from 4 mice in Vehicle-ACSF, $n = 7$ cells from four mice in Vehicle-KN93, $n = 7$ cells from four mice in Ketamine-ACSF and $n = 8$ cells from 4 mice in Ketamine-KN93). **m** Representative western blot image of Co-IP assay. **n** Quantification of the association between PSD-95 and TARP-γ8 ($n = 7$ in Tat-Scrambled, $n = 8$ in Tat-TTPV). **o** Preference for sucrose in the SPT ($n = 9, 9, 8$ and 8 in Vehicle-Tat-Scrambled, Vehicle-Tat-TTPV, Ketamine-Tat-Scrambled and Ketamine-Tat-TTPV, respectively). **p, q** Immobility time in the FST (**p**) and total distance in the OFT (**q**) ($n = 10, 9, 8$ and 8 in Vehicle-Tat-Scrambled, Vehicle-Tat-TTPV, Ketamine-Tat-Scrambled and Ketamine-Tat-TTPV, respectively). Comparisons were performed by unpaired, two-tailed $t$ test in (**b**, **n**) and two-way ANOVA analysis followed by Bonferroni's multiple comparisons test in (**c**–**l**, **o**–**q**). Data are presented as mean ± SEM. All numbers ($n$) are biologically independent experiments. Source data are provided as a Source Data file.

indispensable role of TARP-γ8-PSD-95 coupling in the rapid anti-depressant action of ketamine. Together, these results suggest that CaMKIIα-mediated increase in the binding of TARP-γ8 to PSD-95 plays an essential role in the behavioral benefits of ketamine.

## Specific knockdown of TARP-γ8 in excitatory neurons of ventral hippocampus blocks the rapid antidepressant action of ketamine

Previous report has identified that TARP-γ8 is distributed in different types of neurons throughout the hippocampus[34]. By employing specific antibody for CaMKIIα or GAD67 in immunofluorescent experiments, it was found that TARP-γ8 was mainly co-localized with CaMKIIα rather than GAD67 in the ventral hippocampus (Fig. 6a), indicating that TARP-γ8 is mainly distributed in excitatory neurons. Since the antidepressant actions of ketamine may rely on cell-specific response[35–37], for example, ketamine produces sustained antidepressant action via up-regulation of voltage-gated potassium channel subfamily Q member 2 (*Kcnq2*) gene in the glutamatergic neurons of ventral hippocampus of mice[37], we wondered whether ketamine exerted rapid antidepressant effects via specific regulation of TARP-γ8 in excitatory neuron. To address this issue, we constructed AAV vector containing shRNA targeted for TARP-γ8 with specific CaMKIIα-promoter (*Cacng8*-shRNA) to knockdown TARP-γ8 expression in the excitatory neurons of ventral hippocampus (Fig. 6b). AAV vector-driven specific expression of green fluorescent proteins in the excitatory neuron of ventral hippocampus was verified via immuno-fluorescence using an anti-CaMKIIα antibody (Supplementary Fig. 12a).

We next examined the effect of excitatory neuron-specific knockdown of TARP-γ8 on the antidepressant effects of ketamine. AAV vector (2 μl, each side) was bilaterally infused into the ventral hippocampus of male mice. After 28 days, ketamine (10 mg/kg, i.p.) administration and behavioral tests for 2 h and 24 h were employed. It was shown that knockdown of TARP-γ8 in excitatory neurons was sufficient to induce depressive-like behaviors and blocked the rapid antidepressant effects of ketamine in the SPT and FST, without alterations on locomotion activity (Fig. 6c–f and Supplementary Fig. 12b–d). Moreover, the effect of ketamine on the enhancement of excitatory synaptic transmission was prevented by knockdown of TARP-γ8 in excitatory neurons (Fig. 6g, h). Meanwhile, kinetic properties of AMPARs-mediated mEPSCs were unchanged (Supplementary Fig. 13a, b). These results highlight that TARP-γ8 in the ventral hippocampal excitatory neurons is essential for ketamine-mediated antidepressant effects.

## Discussion

Our present study demonstrated that ketamine treatment facilitated recruitment of TARP-γ8 at the postsynaptic sites and enhanced TARP-γ8-selective AMPARs-mediated synaptic transmission in the ventral hippocampus of male mice, which contributed to the rapid antidepressant effects of ketamine (Fig. 7). TARP-γ8, especially in excitatory neurons of the ventral hippocampus, was indispensable for the regulation of AMPAR signaling and antidepressant action of ketamine.

In recent years, the molecular mechanisms involved in the antidepressant action of ketamine is being identified[38]. The influence of ketamine on the AMPAR signaling is attracting more and more attention. Previous studies have reported that ketamine promotes AMPAR function and synaptogenesis through brain-derived neurotrophic factor/TrkB/mammalian target of rapamycin complex 1(mTORC1) signaling pathway[21,39]. These studies focus on the promotion of synthesis of synaptic proteins, including PSD-95 and AMPARs, which are required for the formation of synapse. However, little is known about how ketamine affects dynamic regulation of AMPARs, particularly trafficking and synaptic anchoring. Here, we found that bath incubation of ketamine increased the amplitude of AMPARs-mediated mEPSC in the ventral hippocampal CA1 neurons, indicating a facilitated recruitment of AMPAR complexes into postsynaptic membrane. However, whether ketamine-triggered synaptic localization of AMPAR is dependent on the inhibition of NMDAR is unclear. The "indirect" disinhibition of pyramidal neuron induced by selective blockade of NMDAR on GABAergic interneurons or the inhibition of extrasynaptic NMDAR and subsequent mTORC1 activation indicate the NMDAR-dependent antidepressant actions of ketamine. Nevertheless, Zanos et al showed that ketamine metabolites exerted antidepressant effects independent of inhibition of NMDARs[1]. Knocking out subunits of NMDAR (GluN1 or GluN2C) also does not affect the molecular and behavioral benefit of ketamine[40,41]. In our electrophysiological recordings, mEPSCs were obtained from pyramidal neurons in the ventral hippocampal CA1 region in the presence of D-AP5, which completely blocked NMDAR-mediated currents. Bath incubation of ketamine for 1 h enhanced the amplitude of AMPARs-mediated mEPSC. However, there is still a lack of evidence that ketamine increases AMPAR-mediated currents independent of NMDAR, given that ketamine itself inhibits NMDAR. Moreover, a recent study proposes that indirect activation of the NMDAR subunit GluN2A is necessary to elicit the antidepressant effects of ketamine[42]. Further studies (i.e., knockout of AMPAR or NMDAR subunit in specific brain regions though Cre-Lox system) are need to clarify the roles of AMPARs and NMDARs in the antidepressant actions of ketamine.

A potential involvement of CaMKII signaling in depression has been investigated in human brain postmortem studies in MDD, suggesting the reduced level of CaMKIIα gene in the hippocampus of depressed suicides[43]. Recent reports also show lower expression of CaMKIIβ proteins in the ventral hippocampus of mice following chronic stress[18]. Moreover, previous reports indicate that the antidepressant effect of ketamine requires the activation of CaMKII and increased the synthesis of plasticity-related proteins, such as PSD-95 and GluA1[30]. In our study, bath incubation of KN93 eliminated the enhancement of AMPARs-mediated synaptic transmission induced by ketamine, which further provided evidence that the synaptic action of ketamine is dependent on the activation of CaMKII and synaptic anchoring of AMPAR triggered by ketamine requires CaMKII-dependent TARP-γ8-PSD-95 coupling. Consistent with previous study that phosphorylation of TARP C-terminal by CaMKII facilitates TARP-γ8-PSD-95 interaction[32,33], we found that synaptic localization of TARP-γ8 induced by ketamine was blocked by intra-ventral hippocampus injection of KN93. Moreover, CaMKII inhibition induced by KN93 or disruption of TARP-γ8-PSD-95 binding via Tat-TTPV prevented behavioral responses to ketamine, indicating that TARP-γ8-PSD-95 coupling is necessary for the rapid antidepressant action of ketamine. Recent work demonstrates that CaMKIIβ-dependent TARP-γ8 phosphorylation promotes the expression of AMPAR subunit GluA1 at the post-synaptic sites, which mediates behavioral resilience to chronic stress[18]. However, the mechanism by which phosphorylation of TARP-γ8 leads to increased expression of synaptic AMPAR has not been elucidated. Our study provided evidence that the phosphorylation of CaMKII was involved in the regulation of the binding between TARP-γ8 and PSD-95. Therefore, we speculated that CaMKII-mediated phosphorylation of TARP-γ8 promoted the binding of TARP-γ8 to PSD-95, and the enhanced recruitment of TARP-γ8 at synaptic sites further increased the expression of synaptic AMPAR.

TARP-γ8 controls important aspects of AMPAR trafficking, channel activity and pharmacology in hippocampus[17,44]. Consistent with severe reduction in AMPAR expression found in TARP-γ8[−/−] mice (TARP-γ8 knockout mice)[45], we observed obvious reduction in AMPAR expression after knockdown of TARP-γ8 in the ventral hippocampus. Meanwhile, knockdown of TARP-γ8 in the ventral hippocampus mimicked the depressive-like phenotypes induced by CSDS. However, the paradox is the fact that the total expression of AMPARs was unchanged in stressed mice but decreased in TARP-γ8-knockdown mice. One possible explanation for the difference may be due to the

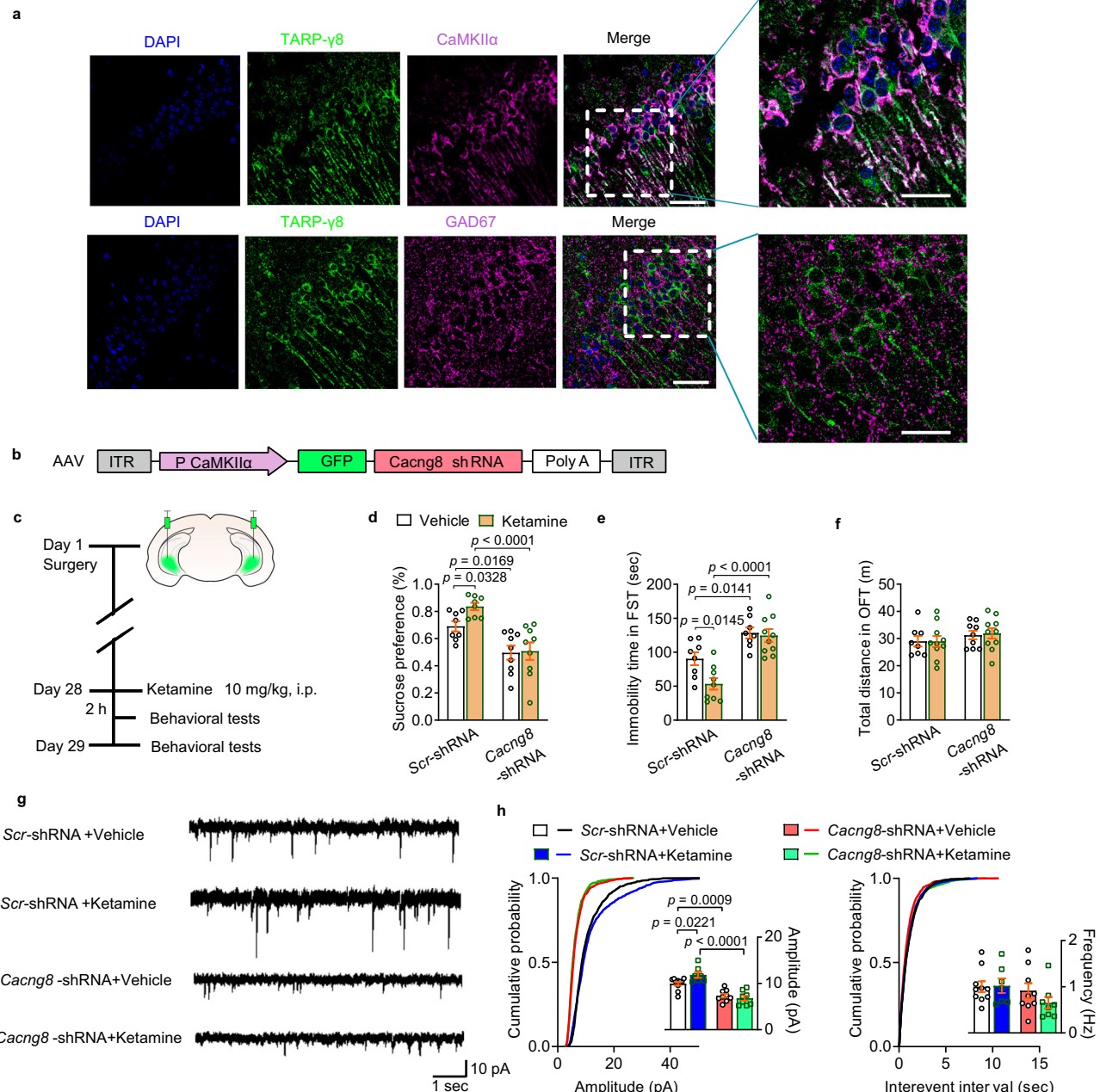

**Fig. 6 | Knockdown of TARP-γ8 in excitatory neurons in the ventral hippocampus blocks the antidepressant effects of ketamine. a** Immunofluorescence results showing TARP-γ8 (green) was mainly expressed in CaMKIIα (purple)-expressing neurons rather than GAD67 (purple)-expressing neurons. Scale bar = 30 μm (two bars on the left), 50 μm (two bars on the right). DAPI (blue). Experiments were repeated independently 3 times with similar results. **b** AAV-expressing constructs with CaMKIIα promoter encoding GFP and shRNAs targeting TARP-γ8. **c** Timeline of experimental procedure. **d** Preference for sucrose in the SPT (*n* = 8 samples per group). **e** Immobility time in the FST (*n* = 8, 8, 9, and 10 in Vehicle-*Scr*-shRNA, Vehicle-*Cacng8*-shRNA, Ketamine-*Scr*-shRNA and Ketamine-*Cacng8*-shRNA, respectively). **f** Total distance in the OFT (*n* = 8, 9, 10 and 10 in

Vehicle-*Scr*-shRNA, Vehicle-*Cacng8*-shRNA, Ketamine-*Scr*-shRNA and Ketamine-*Cacng8*-shRNA, respectively). **g** Representative traces of AMPARs-mediated mEPSC recordings from excitatory neurons in the ventral hippocampus. **h** Quantification of cumulative probability and amplitude and frequency of mEPSCs (*n* = 10 cells from 4 mice in Vehicle-*Scr*-shRNA, *n* = 9 cells from 4 mice in Vehicle-*Cacng8*-shRNA, *n* = 7 cells from 4 mice in Ketamine-*Scr*-shRNA and *n* = 8 cells from 4 mice in Ketamine-*Cacng8*-shRNA). Comparisons were performed by two-way ANOVA analysis followed by Bonferroni's multiple comparisons test. Data are presented as mean ± SEM. All numbers (*n*) are biologically independent experiments. Source data are provided as a Source Data file.

relative specific knockdown of TARP-γ8 by genetic manipulation, but chronic stress-induced loss of TARP-γ8 was only one aspect of homeostatic imbalance under chronic stress condition. A large number of studies have suggested the relatively stable expression of total AMPARs proteins in various stress models[26,46,47]. Thus, we proposed that deficiency of TARP-γ8 diminished the biosynthesis of AMPARs, which may be compensated by other signal pathway in stressed states, leading to unchanged expression of total AMPARs proteins. Assembly

with TARP-γ8 also alters the biophysical properties of AMPARs, reducing speed recovery from desensitization[48]. This compensatory mechanism may also explain the difference in the decay time of mEPSC between TARP-γ8-knockdown and stressed mice.

A recent study shows that, in a BALB/c mouse line, phosphomimic form of TARP-γ8 increased GluA1 expression in the PSD fraction of ventral hippocampus and contributed to behavioral resilience to chronic stress[18]. The study strongly supports our conclusions,

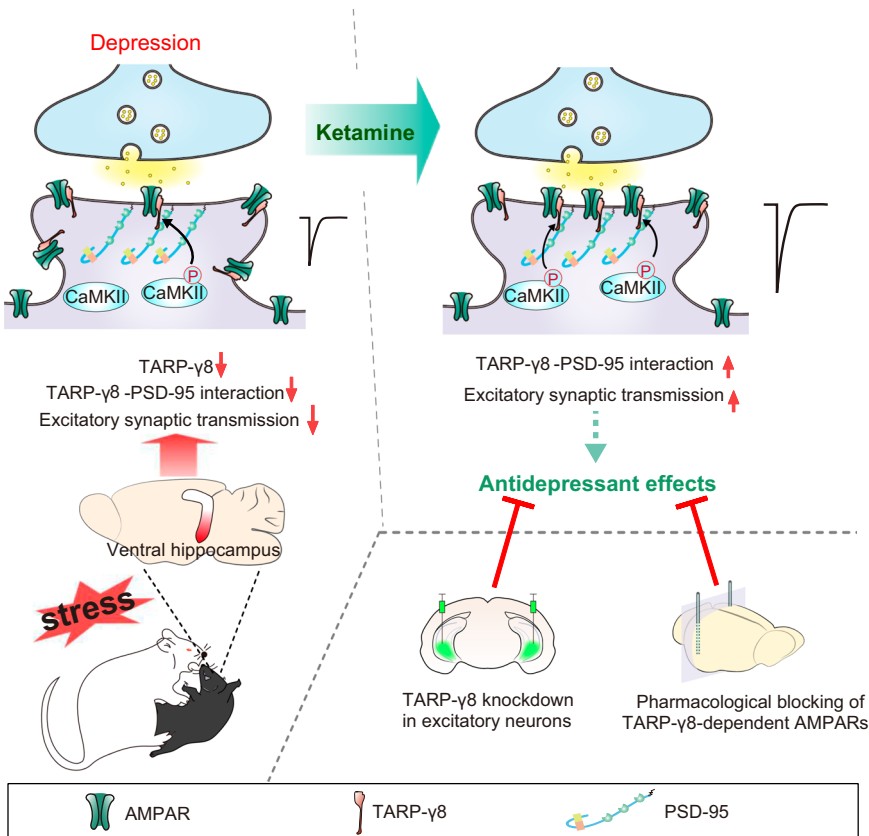

**Fig. 7 | Schematic representation of the suggested model depicting the role of TARP-γ8 in antidepressant effects of ketamine.** Ketamine exhibits anti-depressant effects by increasing the postsynaptic recruitment of TARP-γ8 and the enhanced excitatory synaptic transmission mediated by TARP-γ8-selective AMPAR in a CaMKII phosphorylation-dependent manner in the ventral hippocampus. Knockdown of TARP-γ8 in excitatory neurons of the ventral hippocampus or blockage of TARP-γ8-selective AMPAR abolishes the rapid antidepressant effects of ketamine.

however, a previous study shows that TARP-γ8[-/-] (knockout, KO) mice display decreased immobility time in the forced swim test, which seems contradicting with our findings[27]. An intriguing possibility is the difference in mouse strain. Genetically distinct mouse strains exhibit substantial differences in emotional behavior and stress reactivity. For example, compared with the C57BL/6 stress-resilient strain, BALB/c mice show more susceptible to stressful stimuli[18,49]. Data presented in our study were obtained from C57BL/6 mice. These findings differ from earlier work, where TARP-γ8 KO mice were re-derived in CD1 (ICR) strain, raising the possibility that these two mouse strains show different responses to stress. Furthermore, emotional behavior involves coordinated processing across multiple brain regions[50]. Even the same molecule also has opposite effects on the regulation of emotional behaviors in different brain regions[51,52]. The lack of pro-depressant effect of global TARP-γ8 deletion may be due to opposing effects of TARP-γ8 in different brain regions.

Ketamine act directly on the NMDAR of excitatory neurons, or inhibitory interneurons[53,54]. Recently, cell type-specific molecular mechanism for the antidepressant effects of ketamine has been reported[35–37]. GABAergic interneurons in the mPFC mediates the behavioral response to ketamine in mice, including NMDAR inhibition and increased expression of perineuronal nets in parvalbumin-positive neurons[35,55]. Members of the eukaryotic initiation factor 4E-binding proteins (4E-BPs) family are essential mTORC1 effectors that control mRNA translation and subsequent protein synthesis[56]. Genetic deletion of 4E-BP2 in inhibitory neurons reduces immobility time in the FST, mimicking an antidepressant effect. Ketamine-induced enhancement of excitatory neurotransmission in hippocampal CA1 neurons is also blocked by inhibitory neurons-specific knockout of 4E-BP2[36].

These findings support the idea that GABAergic interneurons in hippocampus are also important for the response to antidepressant activity of ketamine. In the hippocampus, excitatory neurons constitute the major cell population, GABAergic interneurons only account for ~10% of the total neuronal cell population in CA1 region[57,58]. Recently, Lopez et al. identified cell-type-specific molecular signatures associated with ketamine treatment in the ventral hippocampus with single-cell RNA sequencing technique. They demonstrated that *Kcnq2* gene in specific glutamatergic neurons of the ventral hippocampus is essential for the sustained antidepressant-like effects of ketamine[37]. In the present study, we found that TARP-γ8 was predominantly expressed in excitatory neurons and crucial for antidepressant action of ketamine, thus identified a cell type-specific molecular mechanism that mediates the rapid antidepressant actions of ketamine. Future studies will be needed to further accurately quantify the change in surface AMPARs and the interaction between TARP-γ8 and PSD-95 in the excitatory neuron after ketamine treatment. It is worth noting that the clinical manifestations and incidence of depression differ in genders, such as women show a two-fold greater risk compared with men. Females also respond differentially to mechanistically divergent anti-depressants compared with males including ketamine[59]. Further study is needed to investigate in female subjects.

A series of TARP-γ8 antagonists have been designed and synthesized, such as LY3130481 and JNJ55511118, which selectively antagonize AMPAR containing TARP-γ8, but not other TARP members[24,25,60]. Corresponding to this selective activity, LY3130481 was proved to prevent multiple epilepsy-related symptoms in mice without motor side effects[60]. Previous studies demonstrate that the inhibition of global AMPAR blocks the antidepressant effect of ketamine. Here, by use of

JNJ55511118, we provided evidence that specific inhibition of TARP-γ8-AMPAR complex in the hippocampus was sufficient to abolish the antidepressant effect of ketamine. Besides, JNJ55511118 blocked the enhanced excitatory synaptic transmission induced by ketamine in the ventral hippocampus, providing a perspective that TARP-γ8 is involved in the effects of ketamine. In addition, we noticed that JNJ55511118 decreased both amplitude and frequency of mEPSC in the ventral hippocampal CA1 neuron. As a negative modulator of TARP-γ8-containing AMPARs, JNJ55511118 shows high selectivity for TARP-γ8-containing AMPARs and partially inhibits both peak and steady-state glutamate-mediated currents[23]. Therefore, it is expected that JNJ55511118 reduced the amplitude of mEPSC. The frequency of mEPSC is often interpreted as release probability that may be dependent on activity-dependent changes in the numbers of active zones[61]. The reduced mEPSC frequency after JNJ55511118 treatment may reflect the impairment of pre synaptic glutamate release probability. Presynaptic CA3 neurons send Schaffer collaterals to postsynaptic target neurons in CA1, and bath incubation of JNJ55511118 inhibited TARP-γ8-containing AMPARs in the whole hippocampus slices, including CA1 and CA3 subregion. Therefore, JNJ55511118 may decrease the excitability of CA3 neurons through inhibition of TARP-γ8-containing AMPARs, followed by attenuating neurotransmitter release of CA3→CA1, then further resulted in the decreased frequency of mEPSC in CA1 region.

Since our study highlights the important role of TARP-γ8 in antidepressant effects, we support the view that targeting modulation of certain TARPs-containing AMPAR may enable selective modulation of related synaptic transmission in specific brain regions without extensive effect. Recent cryo-electron microscopy works have made great progress in revealing precise stoichiometry and architecture of TARP-γ8-AMPAR complex and the binding site of JNJ5551118[62,63]. AMPARs-related modulators (such as AMPAkine) have been considered as potential targets for drug discovery in MDD[4]. Given the revealed architecture of AMPAR-auxiliary subunit complexes, structure-guided medicinal chemistry of macromolecular receptor targets may promote the development of TARP-γ8-selective AMPAR agonists. Thus, one important future direction of antidepressants may be the development of a selective TARP-γ8 activator to gain region-specific therapeutic intervention for depression. In conclusion, the results of the current study unravel TARP-γ8 as a potential therapeutic target for depression and propose an unidentified mechanism of ketamine's antidepressant effect.

## Methods

### Animals
Adult male C57BL/6 J mice (7 weeks of age, 18–22 g) were obtained from Hunan SJA Laboratory Animal Co., Ltd (Changsha, Hunan, China). Male CD1 mice (<4 months of age) provided by Beijing Vital River Laboratory Animal Technology Co., Ltd (Beijing, China) were retired breeders and singly housed throughout. All animals were housed under 12 h:12 h light/dark cycle with consistent ambient temperature (21–23 °C) and humidity (50% ± 5%) and administered food and water ad libitum. All animal procedures were in accordance with the Institutional Animal Care and Use Committee of Huazhong University of Science and Technology.

### CSDS procedures
CSDS was utilized to mimic features of major depression[19]. C57BL/6 J mice were subjected to social stress for 5–10 min each day, which lasted for 10 days. CD1 mouse was raised on one side of a cage with a perforated Plexiglas divider in the middle. C57BL/6 J mice aged 8 weeks were randomly divided into CSDS group and control group. On the first day, C57BL/6 J mice were placed directly on the side of CD1 mice and subjected to social stress for 5–10 min. During stress time, CD1 mice sniffed and attacked C57BL/6 J mice. After the stress, C57BL/

6 J mice were transferred across the perforated divider to the opposite side for sensory stress for 24 h. Each CD1 mouse was housed in a fixed cage throughout the process, and C57BL/6 J mouse was exposed to a novel CD1 mouse for subsequent daily defeat. After the last day of stress, all C57BL/6 J mice were singly housed in standard mouse cages with ad libitum access to food and water.

### Behavioral assessments
**SIT.** All C57BL/6 J mice were housed singly in standard mouse cages with ad libitum access to food and water. SIT was carried out 24 h after the last defeat session or as required by experimental design[19]. Before testing, C57BL/6 J mice were habituated in the testing room under dim red-light for 1 h. The SIT for each mouse was composed of two 150 s trials: the test mouse was put into the center of an open-field opaque Plexiglas arena (420 mm (w) × 420 mm (d) × 420 mm (h)) with a removable wire-mesh enclosure (100 mm (w) × 65 mm (d)), which did not include a CD1 aggressor mouse (No target) for the first trial and include a novel CD1 aggressor mouse (Target) for the second trial. The whole open-field arena included an "interaction zone" (projecting 8 cm around the wire-mesh enclosure) and two "corner zones" (9 cm × 9 cm area projecting from both corner joints opposing the wire-mesh enclosure). ANY-maze software (Ver 7.1, Stoelting Co., Wood Dale, IL, USA) was used to monitor and record the movement track of mice. The social interaction ratio was obtained by dividing the time spent in the interaction zone when the target was present by when the target was absent, susceptibility to stress was characterized by a reduction in social interaction ratio to values below 1.0.

**SPT.** SPT was used for assessing anhedonia in mice. Mice were single housed in their cages and given 48 h of continuous exposure to two bottles, one containing sucrose water (1% (wt/vol)) and one containing tap water, switching the bottle positions after the first 24 h. Then mice were water deprived for 24 h followed by sucrose preference testing for 2 h during which mice were exposure to two bottles, one containing sucrose water (1% (wt/vol)) and one containing tap water in the dark phase, switching the bottle positions after the first 1 h. Total fluid intake of sucrose water and tap water was recorded. Sucrose preference ratio (%) was calculated as sucrose water consumed/(sucrose water consumed + tap water consumed).

**TST.** TST, based on a previously described procedure[64], was used to evaluate despair-like behavior in mice. First, C57BL/6 J mice were habituated in the testing room under dim red-light for 1 h. Using tape to adhere securely to both the mouse's tail and the suspension bar, mouse was suspended 20 cm above the floor. Video recording device was used to record the video of each mouse testing which usually last for typically 6-min. The time that each mouse spent as no body movement and hung passively was measured as immobile time.

**FST.** FST was carried out to assess the behavioral despair. The procedure was conducted as previously described[65]. C57BL/6 J mice were habituated in the testing room under dim red-light for 1 h. Then, mice were placed individually into a transparent cylindrical glass container measuring 30 cm in height and a diameter of 20 cm containing 20 cm depth water (25 ± 1 °C), the containers were separated from one another using a non-transparent screen in order that the animals would not see each other during testing. The time mice spent floating with the absence of any movement was quantified as the immobility time during the last 4 min of the 6-min test session.

**OFT.** OFT was used to establish the baseline level of motor activity in mice. C57BL/6 J mice were habituated in the testing room under dim red-light for 1 h. Then, mice were placed in an open field Plexiglas box (42 cm (w) × 42 cm (d) × 42 cm (h)) for 10 min, the total distance traveled of mice was recorded using ANY-maze software.

## Electrophysiological recording

For whole-cell patch-clamp recording of mEPSC from CA1 neurons[26,66], mice were anesthetized with isoflurane followed by perfusion with ice-cold oxygenated dissection buffer that contained (in mM): 210 sucrose, 3.1 sodium pyruvate, 11.6 sodium L-ascorbate, 1.0 $NaH_2PO_4$, 26.2 $NaHCO_3$, 5.0 $MgCl_2$ and 20.0 glucose, pH 7.4. Brains were rapidly removed and coronal hippocampal slices (300 µm thick) were cut using a VT-1000S vibratome (Leica, Wetzlar, Germany). Hippocampal slices were incubated in artificial cerebrospinal fluid (ACSF) consisted of (in mM): 119.0 NaCl, 3.5 KCl, 1.3 $MgSO_4$, 2.5 $CaCl_2$, 1.0 $NaH_2PO_4$, 26.2 $NaHCO_3$ and 11.0 glucose, pH 7.4 (300 mOsm) at 28 °C for 1.5 h. After that, brain slices were transferred to a recording chamber mounted on an upright microscope (BX51WIF, Olympus, Tokyo, Japan). For AMPARs-mediated mEPSC recording, patch electrodes (4-6 MΩ) were filled with internal solution containing (in mM): 122.5 Cs-gluconate, 17.5 CsCl, 10.0 HEPES, 0.2 EGTA, 1.0 $MgCl_2$, 4.0 Mg-ATP, 0.3 Na-GTP and 5.0 QX314, (pH 7.2, 280-300 mOsm). mEPSCs recordings were obtained from the ventral hippocampal CA1 pyramidal neurons and recorded at holding potential of −70 mV with a Multiclamp 700B amplifier (Molecular Devices, Sunnyvale, CA, USA) in the presence of bicuculline (20 µM), D-AP5 (50 µM) and Tetrodotoxin (1 µM). Data was analyzed by Axon pClamp10.3 and filtered at 1 kHz and sampled at 10 kHz.

## Western blot analysis

RIPA Lysis Buffer (50 mM Tris (pH 7.4), 150 mM NaCl, 1% Triton X-100, 1% sodium deoxycholate, 0.1% sodium dodecyl sulfate) containing 1% protease inhibitor (Roche, Basel, Switzerland) and phosphatase inhibitor (Sigma-Aldrich, St. Louis, MO, USA) was used for rapid and efficient ventral hippocampus brain tissue lysis and solubilization of proteins. The protein supernatant was collected and quantified with the bicinchoninic acid assay (Beyotime Biotechnology, Haimen, China), then the protein supernatant was mixed with 4× loading buffer and deactivated in 95 °C for 5 min. After that, SDS-PAGE (sodium dodecyl sulfate–polyacrylamide gel electrophoresis), an electrophoretic system, was used to separate proteins with different molecular masses which were then transferred to nitrocellulose membranes. Primary antibodies used to recognize specific protein, including: TARP-γ8 (1:1000; sc-514421, Santa Cruz Biotechnology, Santa Cruz, CA), GluA1 (1:1000; ab31232, Abcam, Cambridge, UK), GluA2 (1:1000; ab52932, Abcam, Cambridge, UK), β-actin (1:3000; sc-47778, Santa Cruz Biotechnology, Santa Cruz, CA), PSD-95 (1:1000; ab18258, Abcam, Cambridge, UK), CaMKIIα (1:800; 50049, Cell Signaling, USA), Phospho-CaMKII (1:800; 12716, Cell Signaling, USA). Secondary antibodies matched for the primary antibody including: IRDye® 800CW Goat anti-Mouse IgG Secondary Antibody (1: 10000; 926-32210; LI-COR Biosciences, USA) and IRDye®800CW Goat anti-Rabbit IgG Secondary Antibody (1: 10000; 926-32211; LI-COR Biosciences, USA). Western blot images were captured with Li-Cor Odyssey (CLx) imaging system and were analyzed using Image Studio Lite (Ver 5.2) software. The information of antibodies used in this study is provided in the Reporting summary.

## Co-immunoprecipitation assay

Co-immunoprecipitation was used to study protein-protein (TARP-γ8/PSD-95) interactions. Ventral hippocampal tissue was lysed in Nonidet P-40 (NP-40) buffer containing 50 mM Tris (pH 7.4), 150 mM NaCl, 1% NP-40 and protease inhibitor cocktail. 300 µg (as output) of total protein was incubated with rabbit anti-PSD-95 antibody or control IgG from rabbit serum overnight at 4 °C. Then the complexes containing the target protein and the interacting proteins were immunoprecipitated with Protein A/G agarose beads (sc-2003, Santa Cruz Biotechnology, Dallas, TX, USA) by centrifugation. The complexes were resuspended in 2× SDS sample buffer and boiled for 5 min at 95 °C.

Proteins were detected by western blot using antibodies specific to the different components.

## Cell-surface biotinylation

Cell-surface biotinylation was used for isolation of plasma membrane proteins. Brain slices containing ventral hippocampus region were washed with ice-cold ACSF. After that, the appropriate volume of 1 mg/ml EZ-link Sulfo-NHS-LC Biotin reagent (21335, ThermoFisher Scientific, Rockford, USA) solution was added to the brain tissue and incubated on ice for 2 h. Brain tissue were washed three times with ACSF plus 100 mM glycine to quench and remove excess biotin reagent. Brain tissue were homogenized with 10 µl /mg RIPA buffer including protease inhibitors cocktail and quantified with bicinchoninic acid assay method. 300 µg of total biotinylated proteins was incubated overnight with Pierce Streptavidin Agarose (20353, ThermoFisher Scientific, Rockford, USA) at 4 °C, streptavidin-biotin complexes were washed three times with RIPA buffer and centrifuged. The complexes were resuspended in 2× SDS sample buffer and boiled for 5 min at 95 °C and detected by western blot.

## Immunohistochemistry analyses

After anesthesia with intraperitoneal sodium pentobarbital (45 mg/kg), mice underwent perfusion intracardially with 0.9% saline followed by 4% paraformaldehyde in PBS. The brain tissues were postfixed in 4% paraformaldehyde at 4 °C for 16 h followed by incubation with 10%-30% sucrose gradient for 2 days at 4 °C. Brain tissues were embedded in OCT and slices containing ventral hippocampus region (30 µm) were obtained by freezing microtome (CM1900, Leica Microsystem, Wetzlar, Germany).

For Nissl staining, the brain slices were stained with Nissl Staining Solution (C0117, Beyotime Inc., Shanghai, China) at 37 °C for 10 min.

For immunofluorescence, brain slices were washed with PBS and incubated with blocking buffer containing 0.1% Triton X-100 and 3% bovine serum albumin (BSA) for 60 min. After blocking, the slides were washed with PBS. Then, the slides were incubated with the primary antibody against TARP-γ8 (1:200; PA5-48249, ThermoFisher Scientific, USA), CaMKIIα (1:200; 50049, Cell Signaling Technology, USA), GAD67 (1:200; ab26116, abcam, Cambridge, UK), overnight at 4 °C. Next, the slices were incubated with Donkey anti-Mouse IgG (H + L) Highly Cross-Adsorbed Secondary Antibody (Alexa Fluor™ 594, 1: 800; A-21203; ThermoFisher Scientific, USA) or Donkey anti-Rabbit IgG (H + L) Highly Cross-Adsorbed Secondary Antibody (Alexa Fluor™ 488, 1: 800; A-21206; ThermoFisher Scientific, USA) which matched for the primary antibody for about 1 h-in the dark followed by DAPI (1:10000; BS130A, Biosharp, China) staining for 5 min. Confocal laser scanning microscope (FV1000, Olympus, Tokyo, Japan) was used for brain slice imaging. The information of antibodies used in this study is provided in the Reporting summary.

## Virus

AAV-U6-*Cacng8*-shRNA-CMV-GFP-pA (Serotype 2/9, titer $7.63 \times 10^{12}$ vg/mL, vector genome per mL) and AAV-U6-scr-shRNA-CMV-EGFP-pA (Serotype 2/9, titer $5.61 \times 10^{12}$ vg/mL) were purchased from BrainVTA, Wuhan. AAV-CaMKIIα-GFP-*Cacng8*-RNAi-PolyA (Serotype 9, titer $3.29 \times 10^{12}$ vg/mL) and AAV-CaMKIIα-GFP-scr-RNAi-PolyA (Serotype 9, titer $4.01 \times 10^{12}$ vg/mL) were purchased from Genechem, Shanghai. LV-CMV-*Cacng8*-IRES-GFP (titer $5 \times 10^8$ TU/mL, transduction unit per mL) were purchased from Integrated Biotech Solutions, Shanghai. LV-CMV-*Cacng8*-Δ4-IRES-GFP (titer $3.5 \times 10^8$ TU/mL) and LV-CMV-scr-IRES-GFP (titer $5 \times 10^8$ TU/mL) were purchased from Genechem, Shanghai.

## Drugs

Ketamine (The third Research Institute of the Ministry of Public Security, Shanghai, China) was diluted in 0.9% sterile saline, KN93 (S7423, Selleck chemicals, USA) and JNJ55511118 (SML1747,

Sigma-Aldrich, St. Louis, MO, USA) were dissolved in DMSO and diluted in ACSF. Tat-TTPV was constituted of TTPV PDZ domain site of TARP-γ8 (LNRKTTPV), and was fused to a human immunodeficient virus (HIV)-1 Tat peptide (YGRKKRRQRRR) in order to permeate cells. The scrambled peptide, Tat-scrambled, was comprised of the same 9 amino acids placed in random sequence (PKTRLNVT). Both Tat-TTPV and Tat-scrambled peptide (GL Biochem Ltd, Shanghai, China) were diluted in ACSF.

## Stereotaxic surgery

Mice were anesthetized with sodium pentobarbital (45 mg/kg, i.p.). The mouse head was put in a fixed position of stereotaxic device, allowing for the precise location of brain area.

For viral injections, adeno-associated virus 2/9 (AAV2/9) vector and Lentivirus vector were bilaterally injected into the ventral hippocampus at the stereotaxic coordinate (-3.08 mm anteroposterior; ± 3.2 mm mediolateral from bregma; −3.7 mm dorsoventral from the brain surface) through a Nanoliter Injector (World Precision Instruments, Sarasota) at injection rate of 100 nl/min and followed by 10 min of retaining to get optimal virus diffusion. Behavior tests commenced 3–4 weeks later.

For drug infusion, mice were implanted bilaterally into ventral hippocampus with 22-gauge stainless steel guide cannulas (RWD Life Science Co., Shenzhen, China) that terminated 0.5 mm above the target location (AP = -3.08 mm, ML = ± 3.2 mm, DV = −3.2 mm). After surgery, mice were individually housed and allowed at least 7 d for recovery from surgery. Drugs were delivered into the ventral hippocampus with the microsyringe pump via the infusion cannula which extended 0.5 mm beyond the top of the target location side of the guide cannula. During the infusion, mice were anesthetized using isoflurane gas. Ketamine, JNJ55511118, KN93 and Tat-TTPV were microinjected at a rate of 400 nl/min followed by 5-min retaining.

## Cell culture

Hippocampus was dissected from 1- to 2-day-old male C57BL/6 J mice, neurons were dissociated by trypsin (12.5 mg/mL, 1004GR025, Biofroxx). 500000 cells/well were plated within a 6-well plate which was pre-coated with Poly-l-lysine. Cells were grown at 37 °C, 5% $CO_2$ in a humidified incubator. At 11-12 d in vitro (DIV), drug treatment was performed.

## Statistics and reproducibility

Number of biological replicates is defined in the legends of the figures. All experiments were repeated independently at least three times. All data were presented as mean ± SEM. Analysis was performed by GraphPad Prism 8.0. Statistical significance was considered as $p < 0.05$, The unpaired Student's $t$ tests, one-way or two-way analysis of variance followed by Dunnett's or Bonferroni's post hoc tests were used appropriately (Supplementary Dataset 1). Specific statistical methods and post hoc tests are described in the relevant figure legends.

## Reporting summary

Further information on research design is available in the Nature Portfolio Reporting Summary linked to this article.

## Data availability

The raw data and blot images generated in this study are provided in the Supplementary Information/Source Data file. Further information generated in this study is available from the corresponding author upon request. Source data are provided in this paper.

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

## Acknowledgements

This work was supported by the Foundation for National Key R&D Program of China (Grant no. 2021ZD0202900 to J.-G.C.), National Natural Science Foundation of China (Grant no. 82130110 to J.-G.C. and Grant no. U21A20363 to F.W.), Innovative Research Groups of National Natural Science Foundation of China (Grant no. 81721005 to J.-G.C. and F.W.), National Natural Science Foundation of China (no. 81971279 to F.W., no. 81973310 to J.-G.C.), and PCSIRT (no. IRT13016 to J.-G.C). We specifically thank Professor Hua Li, Dr. Meng-Zhu Zheng, and Ms. Xiao-Xia Gu for their contributions to the analysis and interpretation of the data. We thank Ms. Chun Peng for her assistance with western blot.

## Author contributions

S.-G.X. performed the main part of the experiments and wrote the manuscript, J.-G.H., L.-L.L., and M.-M.C. performed behavioral tests, S.-J.S. performed the western blot. J.-G.H. took part in writing and revising the manuscript. F.W. and J.-G.C. designed, supervised this research, and revised the manuscript.

## Competing interests

The authors declare no competing interests.
