## [Peer Review File · Nature Communications]

Enhanced TARP- γ 8-PSD-95 coupling in excitatory neurons contributes to the rapid antidepressant-like action of ketamine in male miceREVIEWER COMMENTS

Reviewer #1 (Remarks to the Author):

This article by Xue et al., tested the role of transmembrane AMPAR regulatory protein- $\gamma 8$ in the antidepressant actions of ketamine using animal tests. They showed that the antidepressant-like actions of ketamine in chronically stressed mice depend on the activity of TARP- $\gamma 8$ and that overexpression of these AMPAR regulatory proteins rescues depressive-like behaviors following chronic stress. Knockdown of TARP- $\gamma 8$ specifically in ventral-hippocampal excitatory neurons induced depressive-like endophenotypes and abolished ketamine's antidepressant-like actions.

These data are novel and potentially important to understand the exact mechanism of antidepressant actions of ketamine. However, I do have some concerns that the authors should address prior to publication.

Major concerns:

- The authors only assess behaviour 2 hours after ketamine administration. Rapid-acting antidepressant efficacy in rodents is usually defined as the unique behavioral effects of ketamine 24 hours after administration of the drug. The authors should show that at least some of their findings apply to the sustained antidepressant-relevant actions of ketamine as well.
- The authors used only male mice for their entire study. This should be added as a limitation within the discussion section of the manuscript.
- My major concern is the concentration of ketamine used in the electrophysiology experiments. What is the rationale of using such a high concentration of ketamine? Previous electrophysiology experiments (PMID: 25128848) using 10 and 100 μM of ketamine showed that ketamine at the antidepressant-relevant concentration of 10 μM induced AMPAR-mediated metaplastic effects. However, by using the likely anesthetic-relevant concentration of 100 μM this plasticity action of ketamine was abolished. Other pre-clinical research also indicates that high doses of ketamine (which would induce ketamine concentrations close to 100 μM) do not exert antidepressant-like behavioral actions in rodents (PMIDs: 27067013, 9518601, 32133835, 31760154, 20724638, 20724638, 27144355 and DOI: doi.org/10.1101/254904).
- In general, the authors do not give the rationale for their selected doses/concentrations throughout the manuscript.

Minor concerns:

- The authors should assess at least some of their findings using an anhedonia-related test, for example sucrose preference. TST and OFT are indicators of antidepressant-like efficacy and not so much indexes of depressive-like behaviors. For example the experiments assessing the effects of JNJ55511118.
- What were the lighting conditions during the behavioral testing?

Reviewer #2 (Remarks to the Author):

This work by Xue et al. tests the involvement of TARP/ $\gamma 8$ -associated AMPA receptors in the antidepressant effect of ketamine. The authors show that full length, but not $\Delta 4$ $\gamma 8$ overexpression in the ventral hippocampus reversed the social avoidance and the despair behaviors of stressed mice induced by CSDS, that $\gamma 8$ downregulation in the ventral hippocampus is sufficient to cause depressive-like behaviors and that a specific $\gamma 8$ -AMPA blocker prevents ketamine-induced reduction in the immobility in the FST. The authors show that these changes correlate with mEPSCs amplitude in CA1 and with AMPAR subunit expression. These are relevant findings and the

behavior and electrophysiology data look convincing. However, the study falls short of providing a mechanism whereby ketamine specifically affects AMPAR/TARP synaptic insertion or function.

Ketamine is an NMDAR blocker. NMDAR activity is a key factor determining the synaptic AMPAR content in the context of different plasticity phenomena. As previous evidence for the role of AMPAR in ketamine action, Zanos et al (Nature, 2016) showed that ketamine metabolites potentiated AMPAR responses, independent of NMDAR, although this potential mechanism is not tested in this study. Conceivably, the actions of ketamine described in this paper may be influenced by the indirect action through NMDAR. Therefore, a discussion of the plausible dual effects of ketamine on NMDA and AMPA receptors is warranted, ideally accompanied by additional experimental data on the regulation of the AMPAR and NMDAR components, with electrophysiological recordings in the presence of NMDAR or AMPAR blockers. AMPAR/NMDAR ratios would test that the AMPAR component is selectively affected by ketamine.

The justification for the focus on TARP γ 8-mediated AMPAR trafficking as the molecular target of ketamine is unclear. It is well established that, in CA1 PNs, AMPAR trafficking relies on specific interactions between g8 and PSD95. Recent studies (Zeng et al, Neuron 2019, Watson et al Nat. comms 2021, Ravi et al, Cell reports 2022, among others) show that AMPAR synaptic content largely relies on TARP γ 8-PSD-95 interactions. Therefore, an increased TARP/PSD-95 interaction is expected regardless of the mechanism driving the synaptic incorporation of AMPAR. The authors show that coimmunoprecipitation of TARPs with PSD-95 is decreased by CSDS and that this is prevented by ketamine, but these experiments are not sufficiently convincing (see below). On the contrary, the data with the specific TARP γ 8-AMPA blocker JNJ55511118 is very interesting, and the relevance of the result would be enhanced by testing the effect of JNJ55511118 in CSDS vs control mice.

In Figure 1, the uncropped gels indicate that different gels are used for each of the WBs, which is suboptimal, particularly for in vivo experiments, more prone to variability. How was the 10 h time point chosen? In addition to the in vivo experiment, an in vitro co-IP experiment would confirm the result and allow the identification of the time course better.

Related, in Figure 3g-h: the effects shown in the WBs are dramatic. The uncropped membranes suggest that a different gel was run for each synaptic protein and another gel for β -actin. What is the justification for this? While it may be acceptable to run each of the synaptic proteins in separate gels (they have similar MWs), each of these membranes should have its own loading control, which is perfectly possible due to the difference in MW with beta-actin. In particular, non-specific bands in the CSDS/GFP band look substantially lighter in all gels except for the beta actin, indicating that reduced apparent synaptic protein levels may be due to less protein being loaded there.

I could not find the "supplementary methods and materials" the methods refers to.

The sex of mice is not indicated, besides a single mention in the animal section of the methods.

I could not find the "supplementary information-source data file" the methods refers to.

Reviewer #3 (Remarks to the Author):

The study by Xue and colleagues examines the potential role of TARP-g8-PSD95 in excitatory neurons in the ventral hippocampus in the antidepressant action of ketamine. Previous studies have shown that AMPA receptor antagonists block the antidepressant effects of ketamine. The authors posit that since TARP-g8 is an AMPAR auxiliary subunit involved in AMPAR assembly and trafficking in the hippocampus, it may be involved in the antidepressant effects of ketamine. The study proposes an interesting hypothesis but there are several concerns that should be addressed.

1. The authors show that the total expression of TARP-g8 in the ventral hippocampus is downregulated by chronic social defeat stress (CSDS) but not affected by ketamine. PSD95 expression is not altered by CSDS or ketamine. Therefore, it is unclear what the significant change in TARP-g8/PSD95 (percent of control group) represents in the stress (and ketamine) conditions. This is particularly important as numerous stress studies have not observed a change in AMPAR expression.
2. Why does the negative modulator of AMPARs containing TARP-g8, JNJ55511118, decrease both mEPSC amplitude and frequency? What would be the potential presynaptic effect of this compound?
3. Previous work has shown ketamine increases surface expression of GluA1 and GluA2, that are required for the antidepressant effects. How does TARP-g8-PSD95 interact with GluA1 and GluA2 to trigger their increase in surface expression by ketamine?
4. The molecular mechanism for how ketamine mediates its effects on TARP-g8 remains unclear.
5. Related to point #3, Page 14, it is unclear how ketamine acts on GABAergic interneurons in the mPFC to then lead to trafficking of AMPAR by TARP-g8 in excitatory neurons in the hippocampus. How does this work from a mechanistic standpoint?
6. Page 15, discussion. How would a TARP-g8 selective AMPA agonist represent a possible therapeutic for depression? Results with AMPAkinase in depression have been disappointing. The effects of ketamine have shown a requirement for AMPAR but it appears that AMPA receptor potentiation is downstream of NMDA receptor block.
7. There are numerous places that the authors have cited incorrect studies and several key original papers that initially reported ketamine's effects in the hippocampus have been omitted.

Response to Reviewer #1

This article by Xue et al., tested the role of transmembrane AMPAR regulatory protein- $\gamma 8$ in the antidepressant actions of ketamine using animal tests. They showed that the antidepressant-like actions of ketamine in chronically stressed mice depend on the activity of TARP- $\gamma 8$ and that overexpression of these AMPAR regulatory proteins rescues depressive-like behaviors following chronic stress. Knockdown of TARP- $\gamma 8$ specifically in ventral-hippocampal excitatory neurons induced depressive-like endophenotypes and abolished ketamine's antidepressant-like actions.

These data are novel and potentially important to understand the exact mechanism of antidepressant actions of ketamine. However, I do have some concerns that the authors should address prior to publication.

Response: Thank you for your positive evaluations.

Major concerns:

- *The authors only assess behaviour 2 hours after ketamine administration. Rapid-acting antidepressant efficacy in rodents is usually defined as the unique behavioral effects of ketamine 24 hours after administration of the drug. The authors should show that at least some of their findings apply to the sustained antidepressant-relevant actions of ketamine as well.*

Response: Thank you for your important suggestion. Previous studies have demonstrated that ketamine shows a rapid antidepressant effect within hours. In addition to 24 h after administration of ketamine, the rapid-acting antidepressant efficacy is often defined after administration of ketamine for 1 or 2 h (Zanos et al., 2016; Zhang et al., 2017; Yang et al., 2018; Klein et al., 2020; Aguilar et al., 2021). In our study, we determined the rapid-acting antidepressant efficacy of ketamine at 2 h after administration. We followed the reviewer's suggestion, and performed additional experiments to observe the behavioral effects of ketamine 24 h after administration. The results showed that ketamine exhibited antidepressant effects for at least 24 h in control and stressed mice (see page 5, line 111-118; Supplementary Fig. 2a-e). In addition, knockdown of TARP- $\gamma 8$ in excitatory

neurons abolished the antidepressant effects of ketamine after administration for 2 or 24 h (see page 11, line 296-301; Fig. 5d-f and Supplementary Fig. 11b-d). Furthermore, we also found that the binding of TARP- γ 8 to PSD-95 was increased 24 h after ketamine treatment in primary cultured hippocampal neurons and we have addressed the issues in the revised manuscript (see page 6, line 152-155; Fig. 1k-n).

• *The authors used only male mice for their entire study. This should be added as a limitation within the discussion section of the manuscript*

Response: Thank you for pointing out this problem. According to the standardized protocol for chronic social defeat stress, we used the male mice in this study. We followed your suggestion and added the description that only male mice were used in our study in the title, abstract and results, and discussed the limitation in the revised manuscript (see page 16, line 432-436).

• *My major concern is the concentration of ketamine used in the electrophysiology experiments. What is the rationale of using such a high concentration of ketamine? Previous electrophysiology experiments (PMID: 25128848) using 10 and 100 μ M of ketamine showed that ketamine at the antidepressant-relevant concentration of 10 μ M induced AMPAR-mediated metaplastic effects. However, by using the likely anesthetic-relevant concentration of 100 μ M this plasticity action of ketamine was abolished. Other pre-clinical research also indicates that high doses of ketamine (which would induce ketamine concentrations close to 100 μ M) do not exert antidepressant-like behavioral actions in rodents (PMIDs: 27067013, 9518601, 32133835, 31760154, 20724638, 20724638, 27144355 and DOI: doi.org/10.1101/254904)*

Response: Thank you for your valuable comments. We screened for effective concentrations of ketamine to investigate its effect on mEPSCs of CA1 neurons in ventral hippocampal slices, and compared the effects of different concentrations of ketamine (10 μ M and 100 μ M) on mEPSCs of CA1 neurons in ventral hippocampal slices. It was found that both 10 μ M and 100 μ M ketamine enhanced

the amplitude of mEPSCs in ventral hippocampal slices (see page 6, line 130-134; Supplementary Fig. 4). In other studies, 50 μ M or 100 μ M ketamine that used in electrophysiological recordings has been employed to investigate the mechanisms underlying the antidepressant effects of ketamine (Suzuki et al., 2021; Yang et al., 2018).

Izumi et al reported that effects of low ketamine (1-10 μ M) on somatic EPSPs and LTP in the pyramidal cell body layer (dorsal or ventral CA1 was not indicated) were not mimicked by a 100 μ M ketamine concentration (Izumi et al., 2021). In their electrophysiological experiments, they employed extracellular recordings under electrical stimulation conditions. However, in our study, there was no electrical stimulation during patch-clamp recording in the ventral hippocampus, which may contribute to the discrepancy of ketamine concentration in the electrophysiological recording.

As for why high concentrations of ketamine have no behavioral improvement effect, but can trigger rapid synaptic scaling as low concentrations, we speculate that enhanced TARPs- γ 8-PSD-95 coupling is a necessary but sufficient condition for the antidepressant effect of ketamine. In other words, although high-dose ketamine can trigger rapid synaptic scaling in CA1 neurons in ventral hippocampal slices, intraperitoneal administration of high dose ketamine will also cause other effects (such as complete blockade of NMDAR in the whole brain), which may further counteracts the antidepressant effects of rapid synaptic scaling CA1 neurons.

• In general, the authors do not give the rationale for their selected doses/concentrations throughout the manuscript.

Response: Thank you for your professional reminder. In our study, the selected dose of ketamine (10 mg/kg, i.p.) was referred to previous reports, for example, Zanos et al reported that a single dose of ketamine (10 mg/kg, i.p.) produced rapid antidepressant effects in rodents (Zanos et al., 2016). In electrophysiological experiments, we compared the effects of different concentrations of ketamine (10

μM and $100 \mu\text{M}$) on mEPSCs of CA1 neurons in ventral hippocampal slices. Although $10 \mu\text{M}$ and $100 \mu\text{M}$ concentrations of ketamine enhanced the amplitude of mEPSC *in vitro*, the effect at $100 \mu\text{M}$ was more pronounced. Thus, bath incubation of ketamine ($100 \mu\text{M}$) was used in subsequent electrophysiological experiments (see page 6, line 130-134, Supplementary Fig. 4). For local administration of ketamine into the brain, the selected dose of ketamine was referred to the literatures (Yang et al., 2018; Sakai et al., 2021). In addition, we screened the effective antidepressant dosages of intral-ventral hippocampus injection of ketamine or JNJ55511118 in behavioral test. We observed that a single intral-ventral hippocampus injection of ketamine ($2 \mu\text{g}$, bilaterally) produced antidepressant effects after 1 h and 24 h administration (see page 5, line 122-124, Supplementary Fig. 3), and intra-ventral hippocampus injection of $10 \mu\text{M}$, but not $1 \mu\text{M}$ JNJ55511118 induced depressive-like behaviors in mice (see page 7, line 177-181; Fig. 2b-c and Supplementary Fig. 6).

Minor concerns:

- *The authors should assess at least some of their findings using an anhedonia-related test, for example sucrose preference. TST and OFT are indicators of antidepressant-like efficacy and not so much indexes of depressive-like behaviors. For example the experiments assessing the effects of JNJ55511118.*

Response: It is a good suggestion. We performed additional experiments to assess the results of SPT at 2 h and 24 h after administration of ketamine to the mice that received chronic stress. We found that CSDS induced anhedonia in mice, which was reversed by ketamine administration at different time points (see page 5, line 108-111; Fig. 1f; Supplementary Fig. 2c). We also found that intra-ventral hippocampus injection of $10 \mu\text{M}$ JNJ55511118 or specific knockdown of TARP- $\gamma 8$ in excitatory neurons of ventral hippocampus decreased sucrose preference ratio, which further blocked the antidepressant effects of ketamine for 2 h and 24 h (see page 7, line 177-181; page 11, line 299-301; Fig. 2b, 5d; Supplementary Fig. 6a, 7b and 11b).

- *What were the lighting conditions during the behavioral testing?*

Response: Thank you. In our study, all behavioral tests were conducted under dim red-light conditions in a soundproof room. We added the description in the revised manuscript (see page 19-20, line 506, 530, 537, 544).

Response to Reviewer #2

This work by Xue et al. tests the involvement of TARP/ γ 8-associated AMPA receptors in the antidepressant effect of ketamine. The authors show that full length, but not Δ 4 γ 8 overexpression in the ventral hippocampus reversed the social avoidance and the despair behaviors of stressed mice induced by CSDS, that γ 8 downregulation in the ventral hippocampus is sufficient to cause depressive-like behaviors and that a specific γ 8-AMPA blocker prevents ketamine-induced reduction in the immobility in the FST. The authors show that these changes correlate with mEPSCs amplitude in CA1 and with AMPAR subunit expression. These are relevant findings and the behavior and electrophysiology data look convincing. However, the study falls short of providing a mechanism whereby ketamine specifically affects AMPAR/TARP synaptic insertion or function.

Response: Thank you for your general evaluation and professional comments.

Ketamine is an NMDAR blocker. NMDAR activity is a key factor determining the synaptic AMPAR content in the context of different plasticity phenomena. As previous evidence for the role of AMPAR in ketamine action, Zanos et al (Nature, 2016) showed that ketamine metabolites potentiated AMPAR responses, independent of NMDAR, although this potential mechanism is not tested in this study. Conceivably, the actions of ketamine described in this paper may be influenced by the indirect action through NMDAR. Therefore, a discussion of the plausible dual effects of ketamine on NMDA and AMPA receptors is warranted, ideally accompanied by additional experimental data on the regulation of the AMPAR and NMDAR components, with electrophysiological recordings in the presence of NMDAR or AMPAR blockers.

AMPA/NMDAR ratios would test that the AMPAR component is selectively affected by ketamine.

Response: Thank you for your professional reminder. Whether ketamine exerts antidepressant effects through NMDAR inhibition is controversial. The “indirect” disinhibition of pyramidal neuron induced by selective blockade of NMDAR on GABAergic interneurons or the inhibition of extrasynaptic NMDAR and subsequent mTORC1 activation indicate the NMDAR-dependent antidepressant actions of ketamine. Although Zanos et al showed that ketamine metabolites exerted antidepressant effects independent of inhibition of NMDARs (Zanos et al., 2016), this hypothesis is based on the result that (R)-ketamine has fourfold less affinity for the NMDAR than (S)-ketamine yet showed greater potency and longer-lasting antidepressant. Considering that both (R)-ketamine and (S)-ketamine inhibits NMDAR, this conclusion does not support that the antidepressant effect of ketamine is independent of inhibition of NMDARs. Moreover, NMDAR antagonist MK-801, which binds to the same receptor site as ketamine, exhibits comparable acute (1 h) antidepressant effects that is similar to ketamine. Further study such as knockout of NMDAR in specific brain regions is needed to clarify whether the antidepressant effect of ketamine depends on NMDAR. Our study mainly focused on the regulation of ketamine on TARP- γ 8-selective AMPARs at synaptic site. We followed your suggestion and performed mEPSCs recordings that obtained from the ventral hippocampal CA1 pyramidal neurons in the presence of D-AP5 (50 μ M), which completely blocked NMDAR-mediated currents. Bath incubation of ketamine (100 μ M) for 1 hour enhanced the amplitude of AMPARs-mediated mEPSC. Despite these results, there is still a lack of evidence that ketamine increases AMPAR-mediated currents independent of NMDAR, given that ketamine itself inhibits NMDAR. We have addressed the issues in the revised manuscript (see page 12-13, line 327-343).

In addition, we agree with the reviewer that AMPAR/NMDAR ratio would test the role of AMPAR component in some experiments. AMPAR/NMDAR ratios is often used as a normalization procedure to compare the strength of excitatory

synapses. However, in our experimental conditions, ketamine (100 μ M) completely blocked NMDAR (see below, Figure 1), so this parameter cannot be accurately calculated.

Figure 1. The peak amplitude of EPSC at -70 mV refers to AMPAR-EPSCs, the magnitude of the EPSC recorded at +40 mV at 50 ms after afferent stimulation refers to NMDAR-EPSCs.

The justification for the focus on TARP- γ 8-mediated AMPAR trafficking as the molecular target of ketamine is unclear. It is well established that, in CA1 PNs, AMPAR trafficking relies on specific interactions between γ 8 and PSD95. Recent studies (Zeng et al, Neuron 2019, Watson et al Nat. comms 2021, Ravi et al, Cell reports 2022, among others) show that AMPAR synaptic content largely relies on TARP- γ 8-PSD-95 interactions. Therefore, an increased TARP/PSD-95 interaction is expected regardless of the mechanism driving the synaptic incorporation of AMPAR. The authors show that coimmunoprecipitation of TARPs with PSD-95 is decreased by CSDS and that this is prevented by ketamine, but these experiments are not sufficiently convincing (see below). On the contrary, the data with the specific TARP γ 8-AMPA blocker JNJ55511118 is very interesting, and the relevance of the result would be enhanced by testing the effect of JNJ55511118 in CSDS vs control mice.

Response: Thanks for your professional comments. In our study, we found that ketamine increased the binding of TARP- γ 8 to PSD95, and then mediated the surface trafficking of AMPAR and produces antidepressant effects. Blocking the

combination of TARP- γ 8 and PSD95 through deleting the PDZ domain of TARP- γ 8 failed to increase the surface trafficking of AMPAR and did not produce antidepressant effects. Knockdown of TARP- γ 8 resulted in reduced surface expression of AMPAR and blocked the antidepressant effects of ketamine. We also verified the enhanced effect of ketamine on the binding of TARP- γ 8 to PSD-95 in primary hippocampal neurons (see the following response). Therefore, we proposed that the TARP- γ 8-mediated AMPAR trafficking participated in the mechanism of action of ketamine. However, as the reviewer mentioned, the molecular target is still unclear. Further studies are needed to investigate how ketamine mediated AMPAR trafficking by acting on TARP- γ 8. We have addressed the issues in the revised manuscript (see page 13-14, line 355-368).

Our previous results demonstrated that intral-ventral hippocampus injection of JNJ55511118 induced depressive-like behaviors in naïve mice and is also sufficient to abolish the rapid antidepressant effects of ketamine. Based on these previous results, we further tested effects of JNJ55511118 in the CSDS model according to the reviewer's suggestion. We found that JNJ55511118 further prevent ketamine's rapid antidepressant-like actions for 2 h and 24 h in the CSDS model. These results suggested that the activity of TARP- γ 8-selective AMPAR in the ventral hippocampus plays a key role in mediating the antidepressant effects of ketamine (see page 7-8, line 184-188; Fig. 2g-j; Supplementary Fig. 7).

In Figure 1, the uncropped gels indicate that different gels are used for each of the WBs, which is suboptimal, particularly for in vivo experiments, more prone to variability. How was the 10 h time point chosen? In addition to the in vivo experiment, an in vitro co-IP experiment would confirm the result and allow the identification of the time course better.

Response: Thank you for your professional comments and useful suggestion. Actually, in the data analysis processing, the protein expression with their own loading control on the same piece of gel was used for comparison. However, for representative figure, in order to show the bands more clearly and completely, the

same sample was loaded repeatedly in different gels. We have now improved the key representative diagram of WB, for example, the loading control (β -actin) for TARP- γ 8 and total GluA1, GluA2 proteins have been improved in the revised manuscript (see input in Fig. 1o; Fig. 3g; Fig. 4c; GluA1 (t) and GluA2 (t) in 4g and Uncropped scans of key western blots)

Since during Co-IP processing, PSD-95 antibody was used for PSD-95-TARP- γ 8 complex-immunoprecipitation (IP antibody), two non-specific signal bands, heavy chain (55 kDa) and light chain (25 kDa) were generated after IP antibody denaturation. We found that the heavy chain was dispersed and near the detected target protein (TARP- γ 8) band. Therefore, for representative diagram of Co-IP-output-WB, even same gel was used for sample-protein loaded, we cut apart the whole nitrocellulose membrane and split it in two parts. PSD-95 antibody was incubated in the one nitrocellulose membrane (molecular weight > 70 kDa) and TARP- γ 8 antibody was incubated in the other one (molecular weight < 70 kDa). Related results were presented in Fig. 1k, 1o and Uncropped scans of key western blots.

We are sorry that we did not find 10 h time point in the manuscript. Indeed, we only screened different concentrations of ketamine, not different time points. The antidepressant effect of ketamine was detected 2 h after intraperitoneal injection. Considering that it takes a certain amount of time for ketamine to be absorbed and metabolized in the body and to be detected in the brain, behavioral tests were assessed 1 h after intral-ventral hippocampus injection of ketamine. In addition, Yang et al also showed the antidepressant effects in rats 1 h after administration of ketamine into the lateral habenula (Yang et al., 2018).

We followed the reviewer's suggestion and evaluated the effect of ketamine on the binding of TARP- γ 8 to PSD-95 in primary hippocampal neurons and screened different time point of ketamine treatment. The results showed that ketamine increased the interaction between TARP- γ 8 and PSD-95 in primary hippocampal neurons 1 h or 24 h after treatment (see page 6, line 152-155; Fig. 1k-n).

Related, in Figure 3g-h: the effects shown in the WBs are dramatic. The uncropped membranes suggest that a different gel was run for each synaptic protein and another gel for β -actin. What is the justification for this? While it may be acceptable to run each of the synaptic proteins in separate gels (they have similar MWs), each of these membranes should have its own loading control, which is perfectly possible due to the difference in MW with beta-actin. In particular, non-specific bands in the CSDS/GFP band look substantially lighter in all gels except for the beta actin, indicating that reduced apparent synaptic protein levels may be due to less protein being loaded there.

Response: We apologize for the confusing presentation of uncropped membranes in Figure 3g-h. As the reviewer mentioned, each of the synaptic proteins was presented in separate gels because of the similar MWs, especially for GluA1 and GluA2. We presented the loading control (β -actin) for TARP- γ 8 and total GluA1, GluA2 proteins in the revised manuscript (see Fig. 3g and Uncropped scans of key western blots), except for membrane protein fraction that referred to surface/total expression of protein.

I could not find the “supplementary methods and materials” the methods refers to.

Response: We apologize for that we did not provide a separate file for “supplementary methods and materials”, but integrated into the main text file. Please see this part from Lines 477-675 in the revised version of the manuscript.

The sex of mice is not indicated, besides a single mention in the animal section of the methods.

Response: Thank you. Our study only focused on male mice. We added this information in the title, abstract and results (see page 2, line 31, 33). In addition, we also added a new statement to the discussions to clarify the limitation of lack of female mice in our study (see page 16, line 432-436).

I could not find the “supplementary information-source data file” the methods refers

to.

Response: We have now uploaded “Source data” as a separate file in the revised manuscript.

Response to Reviewer #3

The study by Xue and colleagues examines the potential role of TARP-g8-PSD95 in excitatory neurons in the ventral hippocampus in the antidepressant action of ketamine. Previous studies have shown that AMPA receptor antagonists block the antidepressant effects of ketamine. The authors posit that since TARP-g8 is an AMPAR auxiliary subunit involved in AMPAR assembly and trafficking in the hippocampus, it may be involved in the antidepressant effects of ketamine. The study proposes an interesting hypothesis but there are several concerns that should be addressed.

Response: We thank the reviewer for the positive comments on our work. In the following sections, you will find our responses to each of your points and suggestions.

1. The authors show that the total expression of TARP-g8 in the ventral hippocampus is downregulated by chronic social defeat stress (CSDS) but not affected by ketamine. PSD95 expression is not altered by CSDS or ketamine. Therefore, it is unclear what the significant change in TARP-g8/PSD95 binding (percent of control group) represents in the stress (and ketamine) conditions. This is particularly important as numerous stress studies have not observed a change in AMPAR expression.

Response: We thank the reviewer for pointing out this issue. We apologize for our improper description of Y-axis in Figure 1h. Changing “TARP-γ8/PSD95 binding (percent of control group)” to “relative TARP-γ8/PSD-95 binding” may be easier to understand (change has been made, see Fig. 1p). Because of the unchanged total PSD-95 protein, we employed the same quantitative PSD-95 in co-IP, which can precisely show the changes in TARP-γ8 binding to equal-quality PSD-95 under stress (and ketamine) conditions.

Previous reports have shown that chronic stress does not alter total AMPAR

protein expression, yet decreased expression of surface or synaptosomal GluA1/2 and GluA2 proteins in chronic stress model (Ma et al., 2021; Li et al., 2018; Kallarackal et al., 2013). AMPAR trafficking relies on specific interactions between TARP- γ 8 and PSD-95 in the hippocampus. TARP- γ 8-PSD-95 interaction enhances the surface expression of AMPARs and stabilizes AMPARs at synaptic site. Our study also reported the unchanged expression of total GluA1/2 but the reduced surface expressions of GluA1 and GluA2 in the hippocampus of susceptible mice. The results presented in Figure 3 indicated that CSDS induced deficiency of surface expressions of AMPARs, possibly due to reduced binding of TARP- γ 8 to PSD-95. Although overexpression of both TARP- γ 8 and TARP- γ 8- Δ 4 increased the expression of TARP- γ 8 protein in stressed mice, overexpression of TARP- γ 8- Δ 4 did not increase the membrane expression of AMPAR due to the lack of PZD domain binding to PSD95. Together, these results indicate that the binding of TARP- γ 8/PSD95 plays a key role in the antidepressant effects under stress condition, which have been added in the revised manuscript (see page 9, line 225-232).

2. Why does the negative modulator of AMPARs containing TARP-g8, JNJ55511118, decrease both mEPSC amplitude and frequency? What would be the potential presynaptic effect of this compound?

Response: Thank you for your professional comments. As a negative modulator of TARP- γ 8 containing AMPARs, JNJ55511118 shows high selectivity for TARP- γ 8 containing AMPARs and partially inhibits both peak and steady-state glutamate-mediated currents (Maher et al., 2016; Dohrke et al., 2020). Therefore, it is expected that JNJ55511118 reduced the amplitude of mEPSC. The frequency of mEPSC is often interpreted as release probability that may be dependent on activity- or experience-dependent changes in the numbers of active zones (Zhang et al., 2005). Presynaptic CA3 neurons send Schaffer collaterals to postsynaptic target neurons in CA1, and bath incubation of JNJ55511118 inhibits TARP- γ 8 containing AMPARs in the whole hippocampus slices, including CA1 and CA3

subregion. Therefore, JNJ55511118 may decrease the excitability of CA3 neurons through inhibition of TARP- γ 8 containing AMPARs, followed by attenuating neurotransmitter release of CA3→CA1, then further resulted in the decreased frequency of mEPSC in CA1 region. We also added a new statement to the discussions (see page 17, line 447-461).

3. Previous work has shown ketamine increases surface expression of GluA1 and GluA2, that are required for the antidepressant effects. How does TARP-g8-PSD95 interact with GluA1 and GluA2 to trigger their increase in surface expression by ketamine?

Response: Thank you. It has been demonstrated that in the forebrain neurons, GluA1/2 heteromers containing two TARP- γ 8 subunits at their B/D sites constitute a major AMPAR combination (Schwenk et al., 2014). Deleting the PDZ binding motif of TARP- γ 8 prevents the trafficking of AMPARs to the synapses as well as LTP (Sheng et al., 2018). It is also well established that in CA1 PNs, the interaction between TARP- γ 8 and PSD-95 is crucial and sufficient for synaptic localization of AMPARs (Sheng et al., 2018; Watson et al., 2021; Ravi et al., 2022). In Figure 3, the decreased expression of surface GluA1 and GluA2 proteins induced by CSDS were rescued by overexpression of TARP- γ 8 rather than TARP- γ 8- Δ 4 (lacking the -TTPV PDZ domain) in the ventral hippocampus. These results indicate that TARP- γ 8/PSD95 complex regulates AMPAR trafficking in the CSDS model. Our results showed that ketamine increased the binding of TARP- γ 8 to PSD-95 in vivo and in vitro (see page 6-7, line 152-160). Therefore, ketamine-triggered increase in interaction of TARP- γ 8/PSD-95 is expected to drive the synaptic incorporation and surface expression of AMPAR.

4. The molecular mechanism for how ketamine mediates its effects on TARP-g8 remains unclear.

Response: We thank the reviewer for pointing out this issue. It is not clear how ketamine affects TARP- γ 8 binding to PSD-95. Recently, Castrato et al. provided an important mechanistic insight into interaction of antidepressants and surface

receptors, in which they found that both typical and fast-acting antidepressants directly bound to TrkB, thereby facilitating synaptic localization of TrkB and enhanced the BDNF signaling pathway (Casarotto et al., 2021). Therefore, we hypothesized that direct binding to TARP- γ 8 may be a possible mechanism by which ketamine promotes the TARP- γ 8-PSD-95 coupling. Further detailed studies such as filtration-based radioligand binding and drug docking assays will contribute to indicate the binding of ketamine to TARP- γ 8. Furthermore, TARP-PSD-95 interaction is dynamic and subject to regulation by phosphorylation of serine residues in the TARP intracellular C-terminal domain via an indirect mechanism. Phosphorylation of the TARP C-terminal domain by CaMKII allows binding to PSD-95 and stabilization of receptors at the synapse (Sumioka et al., 2010). Dephosphorylation of these residues by the phosphatase PP1 facilitate the association of the TARP intracellular domain with phospholipids, which in turn disrupts the TARP-PSD-95 interaction (Tomita et al., 2005). Adaikkan et al. showed that ketamine led to differential regulation of CaMKII function, manifested as autoinhibition (pT305 phosphorylation) followed by autoactivation (pT286) of CaMKII α in the hippocampus (Adaikkan et al., 2018). We also hypothesized that ketamine-induced phosphorylation of TARP- γ 8 C-terminal domain by CaMKII may increase the interaction between TARP- γ 8 and PSD-95. We addressed these issues in the revised manuscript (see page 13-14, line 355-368)

5. Related to point #3, Page 14, it is unclear how ketamine acts on GABAergic interneurons in the mPFC to then lead to trafficking of AMPAR by TARP-g8 in excitatory neurons in the hippocampus. How does this work from a mechanistic standpoint?

Response: Thank you for your professional comments. Our study focused on the mechanism of ketamine's rapid antidepressant action in the ventral hippocampus. It is unclear whether GABAergic interneurons in the mPFC are involved in the regulatory process in our study. The disinhibition of pyramidal neuron induced by selective blockade of NMDAR on GABAergic interneurons in the mPFC is an

important hypothesis for ketamine's antidepressant effect (Gerhard et al., 2020). Besides, other signaling pathways underlying ketamine's antidepressant action that do not rely on “disinhibition” have also been extensively reported, such as inhibition of extrasynaptic NMDARs, disinhibition of BDNF translation from eEF2K, activation of mTORC1, and so on (Xu S et al., 2022). Our results suggest that ketamine acts directly on hippocampal neurons to exert antidepressant effects, such as: 1) bath incubation of ketamine enhanced the amplitude of AMPAR-mediated mEPSC, which was prevented by pretreatment with JNJ55511118; 2) ketamine increased the interaction between TARP- γ 8 and PSD-95 in primary hippocampal neurons; 3) Intra-ventral hippocampal injection of ketamine exhibited rapid antidepressant effects. Considering the monosynaptic projection of ventral hippocampal CA1 neurons to mPFC (Gergues et al., 2020), we could not exclude the potential correlation between the effects of ketamine on mPFC and ventral hippocampus, the relevant mechanism is worthy to be explored further.

6. Page 15, discussion. How would a TARP- γ 8 selective AMPA agonist represent a possible therapeutic for depression? Results with AMPAkinetics in depression have been disappointing. The effects of ketamine have shown a requirement for AMPAR but it appears that AMPA receptor potentiation is downstream of NMDA receptor block.

Response: Thank you again for your professional comments. AMPARs are the predominant excitatory neurotransmitter receptors in the brain. However, the majority of available drugs target sequence conserved receptor segments, the ligand-binding domain in the case of positive allosteric modulators or the channel gate region for negative allosteric modulators cause un-wanted side effects due to their ubiquitous presence across the brain, which may lead to the fail of clinical trials of AMPARs-related modulators. Meanwhile, stress induces brain region-specific bidirectional dysregulation of AMPAR-mediated synaptic transmission (He et al., 2023). Therefore, the drugs that directly targets the receptor are lack of specificity in the brain. The diversity of AMPAR auxiliary subunits contribute to the development of region-selective AMPAR therapeutics, such as TARP- γ 8, which

is strongly enriched in the hippocampus. For example, LY-3130481 (CERC-611, ES-481), which selectively antagonizes AMPAR containing TARP- γ 8, shows therapeutic effects in pain and glioma and with Phase-2 clinical trials currently ongoing (<https://clinicaltrials.gov/ct2/show/NCT04714996>). Since our study highlights the important role of TARP- γ 8 in antidepressant effects, we speculate that agonist of AMPAR containing TARP- γ 8 may show region-specific therapeutic intervention of depression.

The “indirect” disinhibition of pyramidal neuron induced by selective blockade of NMDAR on GABAergic interneurons in the mPFC or the inhibition of resting NMDAR and subsequent mTORC1 activation indicate the NMDAR-dependent antidepressant actions of ketamine. However, NMDARs may not be the only target for the antidepressant action of ketamine. Other studies show that ketamine exerts antidepressant effects independent of NMDAR inhibition, but can even directly bind to related receptors or proteins. For example, as mentioned in the previous response to point #4, it is reported that ketamine binds to BDNF receptor TrkB in a stereoselective manner, thereby facilitating surface expression of TrkB and thus promotes BDNF/TrkB signaling-mediated plasticity (Casarotto et al., 2021). Ketamine also inhibits the endocytosis of TrkB by disrupting the interaction between TrkB and AP2M (Fred et al., 2019). Therefore, we proposed that TARP- γ 8-selective AMPAR agonist may represent a possible therapeutic for depression and with fewer side effects.

7. There are numerous places that the authors have cited incorrect studies and several key original papers that initially reported ketamine’s effects in the hippocampus have been omitted.

Response: We are very sorry for the improper citation in the manuscript. We have checked the manuscript to correct some incorrect citation, and added references to the ketamine’s effects in the hippocampus in the revised manuscript (see page 5, line 119).

References

- Adaikkan, C., E. Taha, I. Barrera, O. David and K. Rosenblum (2018). "Calcium/Calmodulin-Dependent Protein Kinase II and Eukaryotic Elongation Factor 2 Kinase Pathways Mediate the Antidepressant Action of Ketamine." *Biol Psychiatry* **84**(1): 65-75.
- Aguilar-Valles, A., *et al.* (2021). "Antidepressant actions of ketamine engage cell-specific translation via eIF4E." *Nature* **590**(7845): 315-319.
- Casarotto, P. C., *et al.* (2021). "Antidepressant drugs act by directly binding to TRKB neurotrophin receptors." *Cell* **184**(5): 1299-1313.
- Dohrke, J. N., J. F. Watson, K. Birchall and I. H. Greger (2020). "Characterizing the binding and function of TARP γ 8-selective AMPA receptor modulators." *J Biol Chem* **295**(43): 14565-14577.
- Fred, S. M., *et al.* (2019). "Pharmacologically diverse antidepressants facilitate TRKB receptor activation by disrupting its interaction with the endocytic adaptor complex AP-2." *J Biol Chem* **294**(48): 18150-18161.
- Gergues, M. M., *et al.* (2020). "Circuit and molecular architecture of a ventral hippocampal network." *Nat Neurosci* **23**(11): 1444-1452.
- Gerhard, D. M., *et al.* (2020). "GABA interneurons are the cellular trigger for ketamine's rapid antidepressant actions." *J Clin Invest* **130**(3): 1336-1349.
- He, J.-G., H.-Y. Zhou, F. Wang and J.-G. Chen (2023). "Dysfunction of Glutamatergic Synaptic Transmission in Depression: Focus on AMPA Receptor Trafficking." *Biological Psychiatry Global Open Science* **3**(2): 187-196.
- Izumi, Y. and C. F. Zorumski (2014). "Metaplastic effects of subanesthetic ketamine on CA1 hippocampal function." *Neuropharmacology* **86**: 273-281.
- Kallarackal, A. J., *et al.* (2013). "Chronic stress induces a selective decrease in AMPA receptor-mediated synaptic excitation at hippocampal temporoammonic-CA1 synapses." *J Neurosci* **33**(40): 15669-15674.
- Klein, M. E., J. Chandra, S. Sheriff and R. Malinow (2020). "Opioid system is necessary but not sufficient for antidepressive actions of ketamine in rodents." *Proc Natl Acad*

- Sci U S A **117**(5): 2656-2662.
- Li, M. X., *et al.* (2018). "Gene deficiency and pharmacological inhibition of caspase-1 confers resilience to chronic social defeat stress via regulating the stability of surface AMPARs." *Mol Psychiatry* **23**(3): 556-568.
- Ma, H., *et al.* (2021). "Amygdala-hippocampal innervation modulates stress-induced depressive-like behaviors through AMPA receptors." *Proc Natl Acad Sci U S A* **118**(6).
- Maher, M. P., *et al.* (2016). "Discovery and Characterization of AMPA Receptor Modulators Selective for TARP- γ 8." *J Pharmacol Exp Ther* **357**(2): 394-414.
- Ravi, A. S., *et al.* (2022). "Long-term potentiation reconstituted with an artificial TARP/PSD-95 complex." *Cell Rep* **41**(2): 111483.
- Sakai, Y., *et al.* (2021). "Gene-environment interactions mediate stress susceptibility and resilience through the CaMKII β /TARP γ -8/AMPA pathway." *iScience* **24**(5): 102504.
- Schwenk, J., *et al.* (2014). "Regional diversity and developmental dynamics of the AMPA-receptor proteome in the mammalian brain." *Neuron* **84**(1): 41-54.
- Sheng, N., *et al.* (2018). "LTP requires postsynaptic PDZ-domain interactions with glutamate receptor/auxiliary protein complexes." *Proc Natl Acad Sci U S A* **115**(15): 3948-3953.
- Sumioka, A., D. Yan and S. Tomita (2010). "TARP phosphorylation regulates synaptic AMPA receptors through lipid bilayers." *Neuron* **66**(5): 755-767.
- Suzuki, K., J. W. Kim, E. Nosyreva, E. T. Kavalali and L. M. Monteggia (2021). "Convergence of distinct signaling pathways on synaptic scaling to trigger rapid antidepressant action." *Cell Rep* **37**(5): 109918.
- Tomita, S., V. Stein, T. J. Stocker, R. A. Nicoll and D. S. Brecht (2005). "Bidirectional synaptic plasticity regulated by phosphorylation of stargazin-like TARPs." *Neuron* **45**(2): 269-277.
- Watson, J. F., A. Pinggera, H. Ho and I. H. Greger (2021). "AMPA receptor anchoring at CA1 synapses is determined by N-terminal domain and TARP γ 8 interactions." *Nat Commun* **12**(1): 5083.

- Xu, S., et al. (2021). "Uncovering the Underlying Mechanisms of Ketamine as a Novel Antidepressant." *Front Pharmacol* 12: 740996.
- Yang, Y., *et al.* (2018). "Ketamine blocks bursting in the lateral habenula to rapidly relieve depression." *Nature* **554**(7692): 317-322.
- Zanos, P., *et al.* (2016). "NMDAR inhibition-independent antidepressant actions of ketamine metabolites." *Nature* **533**(7604): 481-486.
- Zhang, J., Y. Yang, H. Li, J. Cao and L. Xu (2005). "Amplitude/frequency of spontaneous mEPSC correlates to the degree of long-term depression in the CA1 region of the hippocampal slice." *Brain Res* 1050(1-2): 110-117.
- Zhang, K., *et al.* (2016). "Essential roles of AMPA receptor GluA1 phosphorylation and presynaptic HCN channels in fast-acting antidepressant responses of ketamine." *Sci Signal* **9**(458): ra123.

REVIEWER COMMENTS

Reviewer #1 (Remarks to the Author):

The authors' response to my initial comments is commendable. However, I still have reservations regarding the use of a 100 μ M concentration of ketamine in their studies. Although this concentration produced similar results to lower concentrations of ketamine (i.e., 10 μ M, which is considered to be an antidepressant-relevant concentration) on miniature excitatory post-synaptic currents (mEPSCs), I believe it is crucial to address the implications of this finding in the context of in vivo behavioral assessments.

If the observed in vitro effects of ketamine on AMPARs-mediated mEPSC or its effects on the association between PSD-95 AND tarp- γ 8 at 100 μ M represent a critical antidepressant-relevant mechanism of the drug, one would expect that in vivo doses of ketamine ranging from 10 mg/kg to 100 mg/kg would induce comparable antidepressant-like behavioral or biochemical effects in rodents. However, a number of previous studies (PMIDs: 27067013, 9518601, 31760154, 20724638, 27144355), have shown that ketamine at high doses (above 50 mg/kg, corresponding to >50 μ M concentration in the brain) does not exert rapid antidepressant-related actions. Moreover, a newly published study (PMID: 36596696) suggests that ketamine does not induce antidepressant actions at a dose of 100 mg/kg, as it highly inhibits the NMDAR. Instead, this study proposes that indirect activation of the post-synaptic NMDARs is necessary to elicit the antidepressant effects of ketamine. I believe it is essential for the authors to address these discrepancies and provide their explanation for these findings in the Discussion section of their revised manuscript.

By incorporating a thorough discussion of these results and the proposed antidepressant-related mechanism of action of ketamine based on these studies, the authors can further strengthen the scientific rigor and impact of their study.

Reviewer #2 (Remarks to the Author):

In the revised version, the authors substantially improved the study. All my questions and comments have been addressed.

In particular, by testing the effect of hippocampal infusion of JNJ55511118 in the ketamine antidepressant-like actions in the CSDS model, the authors provide important evidence supporting their claim that TARP γ -8 / AMPAR complexes in the ventral hippocampus are key to the antidepressant effects of ketamine.

Minor comments:

- In the Discussion, the authors speculate that CaMKII mediated phosphorylation of the TARP γ -8 cytoplasmic tail upon ketamine treatment might increase TARP-PSD-95 interaction. This is certainly plausible, although there are reports suggesting that these phosphorylations may have an opposite effect in AMPAR/TARP complex synaptic incorporation. As a suggestion, if samples and antibodies (Park et al, 2016) are available, this could be tested by measuring TARP γ -8 cytoplasmic tail S277 phosphorylation in samples treated with ketamine vs. vehicle. To clarify, I do not think this is a requisite for publication but rather a relatively simple and informative experiment if materials are available.

- Please revise AMPAR spelling. Page 13, line 339.

- Please revise CaMKII spelling. Page 13, line 355.

Reviewer #3 (Remarks to the Author):

In this study, the authors present data suggesting that the antidepressant-like actions of ketamine in chronically stressed mice depend on the activity of TARPs and that overexpression of these AMPAR regulatory proteins rescues depressive-like behaviors following chronic stress. The authors attempted to address my earlier questions without providing further data or mechanistic analysis. Unfortunately, several of their answers rely on speculation or a small number of earlier studies, that in face value need to be replicated to establish causality within the context of the current work.

For instance, it remains unclear how ketamine impacts AMPA receptor trafficking and whether the authors' observations are simply due to a "permissive" effect of TARPs on maintenance of AMPA receptors rather than an instructive role of TARPs as a target of ketamine action. The role of AMPA receptor plasticity and trafficking in the field is well-established but a direct instructive role for TARPs, while possible, is not explicitly demonstrated by the current study. In particular, it is essential to link ketamine action (NMDAR block or some other mechanism the authors think is important) directly to TARPs. In the absence of such data, the study is rather correlative. The same concern applies to the role of stress in regulation of AMPAR trafficking, here again, does any of the stress related pathways target TARPs? Or TARPs are "permissively" involved because they are important for AMPAR maintenance?

Response to Reviewer #1

The authors' response to my initial comments is commendable. However, I still have reservations regarding the use of a 100 μ M concentration of ketamine in their studies. Although this concentration produced similar results to lower concentrations of ketamine (i.e., 10 μ M, which is considered to be an antidepressant-relevant concentration) on miniature excitatory post-synaptic currents (mEPSCs), I believe it is crucial to address the implications of this finding in the context of in vivo behavioral assessments.

Response: Thank you for your professional comments. As you mentioned, a number of studies have shown that intraperitoneal injection of high-dose of ketamine (above 50 mg/kg, corresponding to > 50 μ M in the brain) does not produce rapid antidepressant action. Therefore, in our study, low dose of ketamine (10 mg/kg, i.p.) was employed in the behavior tests. Considering that ketamine at concentration of 10 μ M in vitro is considered to be an antidepressant-relevant concentration in vivo (10 mg/kg, i.p), we performed additional experiments to observe the effect of low concentration (10 μ M) of ketamine on mEPSCs in ventral hippocampal slices and primary cultured neurons. We found that ketamine at concentration of 10 μ M and 100 μ M produced similar effects on mEPSCs of CA1 neurons in ventral hippocampus. Therefore, to ensure the consistency between in vivo and in vitro experiments, we only presented the results of low dose of ketamine (10 μ M), but removed the results of high dose of ketamine (100 μ M) in the revised manuscript (see page 6, line 132-135, 146-148; page 7, line 176-183; page 8, line 189-195; page 11, line 290-296; page 13, line 336-339; Fig. 1i-j, 2k-m, 5j-5l, 6j-h; Supplementary Fig. 4, 7, 10 and 13).

If the observed in vitro effects of ketamine on AMPARs-mediated mEPSC or its effects on the association between PSD-95 AND tarp- γ 8 at 100 μ M represent a critical antidepressant-relevant mechanism of the drug, one would expect that in vivo doses of ketamine ranging from 10 mg/kg to 100 mg/kg would induce comparable antidepressant-like behavioral or biochemical effects in rodents. However, a number of

previous studies (PMIDs: 27067013, 9518601, 31760154, 20724638, 27144355), have shown that ketamine at high doses (above 50 mg/kg, corresponding to >50 μ M concentration in the brain) does not exert rapid antidepressant-related actions. Moreover, a newly published study (PMID: 36596696) suggests that ketamine does not induce antidepressant actions at a dose of 100 mg/kg, as it highly inhibits the NMDAR. Instead, this study proposes that indirect activation of the post-synaptic NMDARs is necessary to elicit the antidepressant effects of ketamine. I believe it is essential for the authors to address these discrepancies and provide their explanation for these findings in the Discussion section of their revised manuscript.

By incorporating a thorough discussion of these results and the proposed antidepressant-related mechanism of action of ketamine based on these studies, the authors can further strengthen the scientific rigor and impact of their study.

Response: Thank you for your professional comments. We observed that high concentration of ketamine (100 μ M) *in vitro* triggered rapid synaptic anchoring of AMPAR similar to that of low concentrations of ketamine (10 μ M), suggesting that both high-dose and low-dose of ketamine trigger rapid synaptic scaling in ventral hippocampal CA1 neurons *in vitro*. However, intraperitoneal administration of high dose of ketamine induces other effects, such as complete blockade of NMDAR in the whole brain, which may further counteract the antidepressant effects by rapid synaptic scaling of CA1 neurons. Therefore, we only presented the results of low dose of ketamine (10 μ M), but removed the results of high dose of ketamine (100 μ M) in the revised manuscript. Zanos et al., proposed that indirect activation of the NMDARs (mainly GluN2A) was necessary and sufficient for the antidepressant effects of ketamine, suggesting that different subunits of NMDAR receptor play diverse roles in the antidepressant action of ketamine. Furthermore, the effects of ketamine can be abolished by AMPAR antagonists and mimicked by the AMPARs-positive modulators (Zanos et al., 2016; Gordillo et al., 2020). Further studies (i.e., knockout of AMPAR or NMDAR subunit in specific brain regions though Cre-Lox recombination system) are need to clarify the roles of

AMPARe and NMDARs in the antidepressant actions of ketamine. We addressed this issue in the revised manuscript (see page 14, line 376-378).

Response to Reviewer #2

In the revised version, the authors substantially improved the study. All my questions and comments have been addressed.

In particular, by testing the effect of hippocampal infusion of JNJ55511118 in the ketamine antidepressant-like actions in the CSDS model, the authors provide important evidence supporting their claim that TARP γ -8 / AMPAR complexes in the ventral hippocampus are key to the antidepressant effects of ketamine.

Response: Thank you for your positive evaluation.

Minor comments:

- In the Discussion, the authors speculate that CaMKII mediated phosphorylation of the TARP γ -8 cytoplasmic tail upon ketamine treatment might increase TARP-PSD-95 interaction. This is certainly plausible, although there are reports suggesting that these phosphorylations may have an opposite effect in AMPAR/TARP complex synaptic incorporation. As a suggestion, if samples and antibodies (Park et al, 2016) are available, this could be tested by measuring TARP γ -8 cytoplasmic tail S277 phosphorylation in samples treated with ketamine vs. vehicle. To clarify, I do not think this is a requisite for publication but rather a relatively simple and informative experiment if materials are available.

Response: Thank you for your good suggestion. Although the anti-phospho-TARP- γ 8 S277 antibody is not available, we detected the phosphorylation level of CaMKII α , and found that acute ketamine treatment increased p-CaMKII α level in the ventral hippocampus. Moreover, inhibition of phosphorylation of CaMKII via KN93 prevented molecular and behavioral benefit of ketamine. Despite the lack of direct evidence for the effect of ketamine on TARP- γ 8 phosphorylation, our data provided evidence that CaMKII α mediated the molecular and behavioral

benefit of ketamine, possibly by enhancing the binding of TARP- γ 8 to PSD-95 (see page 10-12, line 271-309; Fig. 5, Supplementary Fig. 10, 11).

- Please revise AMPAR spelling. Page 13, line 339.

Response: We apologize for the wrong spelling. We have changed the misspelling into “AMPAR” (see page 14, line 373).

- Please revise CaMKII spelling. Page 13, line 355.

Response: We apologize for the wrong spelling. We have changed the misspelling to “CaMKII” (see page 14, line 379).

Response to Reviewer #3

In this study, the authors present data suggesting that the antidepressant-like actions of ketamine in chronically stressed mice depend on the activity of TARP- γ 8 and that overexpression of these AMPAR regulatory proteins rescues depressive-like behaviors following chronic stress. The authors attempted to address my earlier questions without providing further data or mechanistic analysis. Unfortunately, several of their answers rely on speculation or a small number of earlier studies, that in face value need to be replicated to establish causality within the context of the current work.

For instance, it remains unclear how ketamine impacts AMPA receptor trafficking and whether the authors' observations are simply due to a “permissive” effect of TARPs on maintenance of AMPA receptors rather an instructive role of TARPs as a target of ketamine action. The role of AMPA receptor plasticity and trafficking in the field is well-established but a direct instructive role for TARPs, while possible, is not explicitly demonstrated by the current study. In particular, it is essential to link ketamine action (NMDAR block or some other mechanism the authors think is important) directly to TARPs. In the absence of such data, the study is rather correlative. The same concern applies to the role of stress in regulation of AMPAR trafficking, here again, does any of the stress related pathways targets TARP? Or TARPs are “permissively” involved because they are important for AMPAR maintenance?

Response: Thank you for your professional comments. We followed your suggestion and performed additional experiments to investigate the molecular mechanism underlying the effect of ketamine on TARP- γ 8-PSD-95 interaction and AMPAR trafficking. Previous study has demonstrated that both typical and fast-acting antidepressants directly bind to TrkB and facilitate synaptic localization of TrkB (Casarotto et al., 2021), suggesting the interaction of antidepressants and membrane receptors. Therefore, we first wondered whether TARP- γ 8, being an AMPA receptor auxiliary subunits, could be a direct pharmacological target of ketamine. Therefore, cellular thermal shift assay (CETSA) and microscale thermophoresis test (MST) were employed.

CETSA is used to study thermal stabilization of proteins upon ligand binding for rapid screening of drug target engagement in living cells (Jafari et al., 2014). In our study, the thermal stability of TARP- γ 8 in HT22 cells was tested within a temperature range of 53 to 63 °C. However, CETSA results showed that ketamine did not affect the thermal stability of TARP- γ 8 proteins within the cells (Fig. a), indicating that TARP- γ 8 proteins is not the binding target of ketamine. Furthermore, MST, an immobilization-free technology for measuring biomolecular interactions, was used. It was found that there was no difference in the temperature-induced change in fluorescence between GFP and TARP- γ 8-GFP group (Fig. b). The above results suggest that ketamine does not directly bind to TARP- γ 8 protein.

(a) CETSA results showed no binding capacity with TARP- γ 8 in HT22 cells treated with 10 μ M ketamine.

(b) MST results showed no direct interaction between GFP-tagged TARP- γ 8 and ketamine (15 min) in lysates from GFP-TARP- γ 8 expressing HEK293T cells (n = 3/group).

Considering that previous studies have shown that phosphorylation of TARP C-terminal by CaMKII facilitates TARP- γ 8-PSD-95 interaction (Sumioka et al., 2010; Opazo et al., 2010), and acute ketamine administration stimulates the phosphorylation of CaMKII, we asked whether the increased phosphorylation of CaMKII and subsequent TARP- γ 8-PSD-95 coupling in the ventral hippocampus mediated the rapid antidepressant effects of ketamine. The results showed that phosphorylated level of CaMKII α was increased in the ventral hippocampus induced by acute treatment with ketamine. Both inhibition of phosphorylation of CaMKII α via KN93 and disruption of TARP- γ 8-PSD-95 binding via Tat-TTPV prevented rapid antidepressant action of ketamine, indicating that the TARP- γ 8-PSD-95 binding regulated by CaMKII phosphorylation contributes to the antidepressant effects of ketamine. KN93 also abolished ketamine-induced potentiation of AMPAR-mediated synaptic transmission in the ventral hippocampus. These results suggest that ketamine-induced recruitment of TARP- γ 8 at the postsynaptic sites is not a permissive action, but mediated by a CaMKII phosphorylation-dependent manner.

Although there was no direct evidence that chronic stress affects AMPAR trafficking through TARPs, previous studies have demonstrated that chronic stress results in decreased phosphorylation of CaMKII in the hippocampus (Sakai et al., 2021; Jiang et al., 2022), which may further reduce the binding of TARP- γ 8 to PSD-95. In addition, Sakai et al. showed that CaMKII β -mediated TARP- γ 8 phosphorylation enhanced the expression of GluA1 in the postsynaptic sites to enable stress resilience (Sakai et al., 2021). Taken together, we speculated that p-CaMKII-mediated phosphorylation of TARP- γ 8 promoted the binding of TARP- γ 8 to PSD-95, and the enhanced recruitment of TARP- γ 8 at synaptic sites further increased the expression of synaptic AMPAR. We addressed these issues in the

revised manuscript (see page 10-12, line 271-309; page 14-15, line 379-404; Fig. 5, Supplementary Fig. 10, 11).

References

1. Casarotto, P. C., *et al.* (2021). "Antidepressant drugs act by directly binding to TRKB neurotrophin receptors." *Cell* **184**(5): 1299-1313.
2. Gordillo-Salas, M., R. Pascual-Antón, J. Ren, J. Greer and A. Adell (2020). "Antidepressant-Like Effects of CX717, a Positive Allosteric Modulator of AMPA Receptors." *Mol Neurobiol* **57**(8): 3498-3507.
3. Jafari, R., *et al.* (2014). "The cellular thermal shift assay for evaluating drug target interactions in cells." *Nat Protoc* **9**(9): 2100-2122.
4. Jiang, X., *et al.* (2022). "Asperosaponin VI ameliorates the CMS-induced depressive-like behaviors by inducing a neuroprotective microglial phenotype in hippocampus via PPAR- γ pathway." *J Neuroinflammation* **19**(1): 115.
5. Opazo, P., *et al.* (2010). "CaMKII triggers the diffusional trapping of surface AMPARs through phosphorylation of stargazin." *Neuron* **67**(2): 239-252.
6. Sakai, Y., *et al.* (2021). "Gene-environment interactions mediate stress susceptibility and resilience through the CaMKII β /TARP γ -8/AMPA pathway." *iScience* **24**: 3-18.
7. Sumioka, A., D. Yan and S. Tomita (2010). "TARP phosphorylation regulates synaptic AMPA receptors through lipid bilayers." *Neuron* **66**(5): 755-767.
8. Zanos, P., *et al.* (2023). "NMDA Receptor Activation-Dependent Antidepressant-Relevant Behavioral and Synaptic Actions of Ketamine." *J Neurosci* **43**(6): 1038-1050.
9. Zanos, P., *et al.* (2016). "NMDAR inhibition-independent antidepressant actions of ketamine metabolites." *Nature* **533**(7604): 481-486.

REVIEWERS' COMMENTS

Reviewer #1 (Remarks to the Author):

The authors have adequately addressed my comments and made significant improvements to their manuscript.

Reviewer #3 (Remarks to the Author):

Unfortunately, the authors responses to my earlier comments raise more questions than answers. First, I do not understand the rationale of assessing whether ketamine directly binds to TARP-g8. The experiment shown in the response is rather tangential and does not necessarily address the specific question although the negative result per se is not surprising. It remains unclear how ketamine impacts AMPA receptor trafficking and whether there is a potential permissive or instructive role of TARPs. Second, the authors now tie their observations to activation of CaMKII, however it remains unclear how CaMKII can be activated by an NMDA receptor antagonist such as ketamine. A vast literature exists showing CaMKII activation requires high calcium levels typically driven by NMDA receptor activation rather than block. Taken together, the findings remain rather inconclusive regarding basic aspects of how ketamine interacts with its targets.

Response to Reviewer #3

Unfortunately, the authors responses to my earlier comments raise more questions than answers. First, I do not understand the rationale of assessing whether ketamine directly binds to TARP-g8. The experiment shown in the response is rather tangential and does not necessarily address the specific question although the negative result per se is not surprising. It remains unclear how ketamine impacts AMPA receptor trafficking and whether there is a potential permissive or instructive role of TARPs.

Response: We thank the reviewer for pointing out this issue. Previous study has reported that most antidepressants including fluoxetine promote TRKB signaling in rodents, which is required for the behavioral effects (Casarotto et al. 2021). Therefore, we asked whether ketamine directly bound to TARP- γ 8, and found that ketamine increased the interaction between TARP- γ 8 and PSD-95, suggesting that ketamine acts as a linker between the TARP- γ 8 and PSD-95 to facilitate their association. Although the molecular docking experiment suggested a potential binding pocket of TARP- γ 8 with ketamine, cellular thermal shift assay (CETSA) and microscale thermophoresis test (MST) showed that ketamine did not directly bind to TARP- γ 8 protein.

Next, we wondered whether ketamine affect the binding of TARP- γ 8 to PSD-95 through certain signaling pathways. It has been reported that acute ketamine administration stimulates the phosphorylation of CaMKII *in vitro* and *in vivo* (Choi et al. 2015, Adaikkan et al. 2018). On the other hand, phosphorylation of TARP C-terminal by CaMKII facilitates TARP- γ 8-PSD-95 interaction and synaptic localization of receptors (Tomita et al. 2005, Sumioka et al. 2010, Hafner et al. 2015). We thus hypothesized that the increased phosphorylation of CaMKII in the ventral hippocampus mediated the molecular and behavioral effects of ketamine. In our study, acute ketamine treatment increased the level of p-CaMKII α in the ventral hippocampus. In addition, inhibition of phosphorylation of CaMKII α via KN93 or disruption of TARP- γ 8-PSD-95 binding via Tat-TTPV prevented rapid antidepressant action of ketamine, indicating that the TARP- γ 8-PSD-95 binding regulated by CaMKII phosphorylation contributes to the

antidepressant effects of ketamine. KN93 also abolished the potentiation of AMPAR-mediated synaptic transmission in the ventral hippocampal slices induced by ketamine. These results suggest that ketamine-triggered TARP- γ 8-PSD-95 coupling is likely not a permissive action, but mediated by a CaMKII phosphorylation-dependent manner.

Second, the authors now tie their observations to activation of CaMKII, however it remains unclear how CaMKII can be activated by an NMDA receptor antagonist such as ketamine. A vast literature exists showing CaMKII activation requires high calcium levels typically driven by NMDA receptor activation rather than block. Taken together, the findings remain rather inconclusive regarding basic aspects of how ketamine interacts with its targets.

Response: Thank you for your professional comments. It is well established that activation of NMDARs with subsequent Ca^{2+} influx in neurons activates CaMKII by phosphorylation at the Thr286, which is seemingly contrary to our results. The reason for this difference may be due to the various effect of ketamine on NMDARs in different types of neurons. Previous study has reported that low dose of ketamine selectively antagonizes NMDARs on inhibitory interneurons, leading to disinhibition and indirect excitation on excitatory pyramidal neurons, which in turn initiates neuronal Ca^{2+} -dependent signaling, protein synthesis and activity-dependent synaptic plasticity (Miller et al. 2016, Widman and McMahon 2018, Gerhard et al. 2020). Therefore, we speculated that ketamine may also increase Ca^{2+} -dependent signaling in excitatory neurons. Although we did not detect the level of CaMKII phosphorylation and TARP- γ 8 expression in excitatory neurons after ketamine administration, however, specific knockdown of TARP- γ 8 in excitatory neurons of ventral hippocampus blocked the rapid antidepressant action of ketamine, indicating that CaMKII α -mediated TARP- γ 8 signaling may mediate the antidepressant effects of ketamine.

In addition, some literature also found that acute ketamine administration stimulates CaMKII phosphorylation in the hippocampus (Choi et al. 2015,

Adaikkan et al. 2018, Abdoulaye et al. 2021). CaMKII inhibitors (TatCN21, KN62 or KN93) block molecular or behavioral benefits of ketamine (Choi et al. 2015, Adaikkan et al. 2018, Tang et al. 2020). Especially, Adaikkan et al. showed that administration of ketamine led to differential regulation of CaMKII α function, manifested as autoinhibition (which lasted 10 to 20 minutes) followed by autoactivation (40 minutes, 60 minutes and 180 minutes after ketamine injection) of CaMKII α in the hippocampus (Adaikkan et al. 2018). In our study, we observed that auto-active phosphorylation of CaMKII α (pT286) was increased 2 hours after ketamine administration, which was consistent with previous reports.

References

- Abdoulaye, I. A., *et al.* (2021). "Ketamine induces lasting antidepressant effects by modulating the NMDAR/CaMKII-mediated synaptic plasticity of the hippocampal dentate gyrus in depressive stroke model." *Neural Plast* **2021**: 6635084.
- Adaikkan, C., E. Taha, I. Barrera, O. David and K. Rosenblum (2018). "Calcium/calmodulin-dependent protein kinase II and eukaryotic elongation factor 2 kinase pathways mediate the antidepressant action of ketamine." *Biol Psychiatry* **84**(1): 65-75.
- Casarotto, P. C., *et al.* (2021). "Antidepressant drugs act by directly binding to TRKB neurotrophin receptors." *Cell* **184**(5): 1299-1313.
- Choi, M., *et al.* (2015). "Ketamine produces antidepressant-like effects through phosphorylation-dependent nuclear export of histone deacetylase 5 (HDAC5) in rats." *Proc Natl Acad Sci U S A* **112**(51): 15755-15760.
- Gerhard, D. M., *et al.* (2020). "GABA interneurons are the cellular trigger for ketamine's rapid antidepressant actions." *J Clin Invest* **130**(3): 1336-1349.
- Hafner, A. S., *et al.* (2015). "Lengthening of the stargazin cytoplasmic tail increases synaptic transmission by promoting interaction to deeper domains of PSD-95." *Neuron* **86**(2): 475-489.

- Miller, O. H., J. T. Moran and B. J. Hall (2016). "Two cellular hypotheses explaining the initiation of ketamine's antidepressant actions: Direct inhibition and disinhibition." *Neuropharmacology* **100**: 17-26.
- Sumioka, A., D. Yan and S. Tomita (2010). "TARP phosphorylation regulates synaptic AMPA receptors through lipid bilayers." *Neuron* **66**(5): 755-767.
- Tang, X. H., *et al.* (2020). "Extrasynaptic CaMKII α is involved in the antidepressant effects of ketamine by downregulating GluN2B receptors in an LPS-induced depression model." *J Neuroinflammation* **17**(1): 181.
- Tomita, S., V. Stein, T. J. Stocker, R. A. Nicoll and D. S. Bredt (2005). "Bidirectional synaptic plasticity regulated by phosphorylation of stargazin-like TARPs." *Neuron* **45**(2): 269-277.
- Widman, A. J. and L. L. McMahon (2018). "Disinhibition of CA1 pyramidal cells by low-dose ketamine and other antagonists with rapid antidepressant efficacy." *Proc Natl Acad Sci U S A* **115**(13): 3007-3016.